# A mevalonate bypass system facilitates elucidation of plastid biology in malaria parasites

**Russell P. Swift**[1], **Krithika Rajaram**[1], **Hans B. Liu**[1], **Amanda Dziedzic**[1], **Anne E. Jedlicka**[1], **Aleah D. Roberts**[1], **Krista A. Matthews**[1¤], **Hugo Jhun**[1], **Namandje N. Bumpus**[2], **Shivendra G. Tewari**[3,4], **Anders Wallqvist**[3], **Sean T. Prigge**[1]*

**1** Department of Molecular Microbiology and Immunology, Johns Hopkins School of Public Health, Baltimore, Maryland, United States of America, **2** Department of Medicine (Division of Clinical Pharmacology), Johns Hopkins University School of Medicine, Baltimore, Maryland, United States of America, **3** Department of Defense Biotechnology High Performance Computing Software Applications Institute, Telemedicine and Advanced Technology Research Center, U.S. Army Medical Research and Development Command, Ft. Detrick, Maryland, United States of America, **4** The Henry M. Jackson Foundation for the Advancement of Military Medicine, Inc., Bethesda, Maryland, United States of America

¤ Current address: Department of Pharmacology, University of Texas Southwestern Medical Center, Dallas, TX, United States of America.

* sprigge2@jh.edu

**Data Availability Statement:** All transcriptomic data have been deposited to the National Center for Biotechnology Information Gene Expression Omnibus repository. The plasmids used in this

## Abstract

Malaria parasites rely on a plastid organelle for survival during the blood stages of infection. However, the entire organelle is dispensable as long as the isoprenoid precursor, isopentenyl pyrophosphate (IPP), is supplemented in the culture medium. We engineered parasites to produce isoprenoid precursors from a mevalonate-dependent pathway, creating a parasite line that replicates normally after the loss of the apicoplast organelle. We show that carbon-labeled mevalonate is specifically incorporated into isoprenoid products, opening new avenues for researching this essential class of metabolites in malaria parasites. We also show that essential apicoplast proteins, such as the enzyme target of the drug fosmidomycin, can be deleted in this mevalonate bypass parasite line, providing a new method to determine the roles of other important apicoplast-resident proteins. Several antibacterial drugs kill malaria parasites by targeting basic processes, such as transcription, in the organelle. We used metabolomic and transcriptomic methods to characterize parasite metabolism after azithromycin treatment triggered loss of the apicoplast and found that parasite metabolism and the production of apicoplast proteins is largely unaltered. These results provide insight into the effects of apicoplast-disrupting drugs, several of which have been used to treat malaria infections in humans. Overall, the mevalonate bypass system provides a way to probe essential aspects of apicoplast biology and study the effects of drugs that target apicoplast processes.

study are available in GenBank under accession numbers MN822297 and MN822298. Transcriptomic data are available at https://www.ncbi.nlm.nih.gov/geo/query/acc.cgi?acc=GSE136169.

**Funding:** This work was supported by the National Institutes of Health R01 AI065853 (STP), the Johns Hopkins Malaria Research Institute, the Bloomberg Family Foundation, and the Network Science Initiative of the U.S. Army Medical Research and Development Command, Ft. Detrick, Maryland (Award W81XWH-15-C-0061; STP). This work was also made possible by UL1 RR025005 from the NIH National Center for Research Resources. KR was supported by the NIH training grant T32AI007417. The opinions and assertions contained herein are the private views of the authors and are not to be construed as official or as reflecting the views of the U.S. Army, the U.S. Department of Defense, or The Henry M. Jackson Foundation for Advancement of Military Medicine, Inc. This paper has been approved for public release with unlimited distribution. The funders had no role in study design, data collection and analysis, decision to publish, or preparation of the manuscript.

**Competing interests:** The authors have declared that no competing interests exist.

## Author summary

Malaria parasites rely on an organelle called the apicoplast for growth and survival. Antimalarial drugs such as azithromycin inhibit basic processes in the apicoplast and result in the disruption of the organelle. Surprisingly, addition of a single metabolite, isopentenyl pyrophosphate (IPP), allows the parasites to survive in culture after disruption of the apicoplast. Unfortunately, using IPP to study this phenomenon has several limitations: IPP is prohibitively expensive, has to be used at high concentrations, and has a half-life less than 5 hours. To address these problems, we engineered parasites to express four enzymes from an alternative pathway capable of producing IPP in the parasites. We validated this new system and used it to metabolically label essential metabolites, to delete an essential apicoplast protein, and to characterize the state of apicoplast-disrupted parasites. A key finding from these studies comes from transcriptomic and metabolomic analysis of parasites treated with the drug azithromycin. We found that apicoplast disruption results in few changes in parasite metabolism. In particular, the expression of hundreds of nuclear-encoded apicoplast proteins are not affected by disruption of the apicoplast organelle, making it likely that apicoplast metabolic pathways and processes are still functional in apicoplast-disrupted parasites.

## Introduction

Malaria parasites contain an essential plastid organelle called the apicoplast. This organelle is believed to have arisen from an endosymbiotic event with an algal species, and contains numerous essential bacterial-like proteins and pathways either not present or highly dissimilar from those found within the human host [1], making it an attractive source of potential drug targets [2,3]. The apicoplast retains its own genome; however, it is highly reduced and provides few clues about apicoplast metabolism. Most proteins that function in the apicoplast are encoded in the parasite nucleus and trafficked to the organelle via an N-terminal bipartite signal comprised of a signal peptide and a transit peptide [4]. It is thought that ~400 proteins, representing ~7% of the parasite genome, are trafficked to the apicoplast [5–8]. Approximately half of the predicted apicoplast-localized proteins have unknown functions [9,10] representing a significant gap in our basic understanding of apicoplast biology.

One of the best studied biosynthetic processes in the apicoplast involves the production of isopentenyl pyrophosphate (IPP), and its isomer dimethylallyl pyrophosphate (DMAPP), via the methylerythritol phosphate (MEP) pathway [11]. IPP and DMAPP serve as precursors for a vast group of biomolecules called isoprenoids, several of which have been studied in malaria parasites [12]. The drug fosmidomycin inhibits this pathway and has been evaluated in human trials as an antimalarial compound [13]. A key insight into the significance of the MEP pathway came in 2011 when it was discovered that the apicoplast organelle can be disrupted in blood-stage parasites (along with loss of its organellar genome) as long as IPP is supplied in sufficient quantities [14]. Presumably, the MEP pathway enzymes (all of which are nuclear-encoded) are no longer able to produce isoprenoid precursors after loss of the apicoplast. It may be that these enzymes are no longer expressed in apicoplast-disrupted parasites, or they may fail to function for some other reason.

The previously established chemical bypass system has proven to be invaluable to the study of the apicoplast in malaria parasites, having been used to interrogate the essentiality of apicoplast specific genes [15–19], screen for apicoplast-specific drugs [15,20–23], and conduct forward genetic screens to elucidate essential apicoplast-specific genes [24]. However, this bypass

system does have drawbacks, which make longer-term, larger scale, and metabolic labeling experiments untenable. These issues include the expense of IPP at ~$25,800/g (Sigma-Aldrich) combined with the high concentration (200μM) required to rescue parasite growth [14], as well as chemical instability of IPP, which has a half-life of 4.5 hours at 37˚C [25], necessitating frequent growth medium exchange. In our hands, we have also experienced incomplete rescue of apicoplast-disrupted parasites to wild-type growth levels, and have had difficulty cryopreserving IPP-dependent parasites. For these reasons, the chemical bypass system is not easily amenable to certain basic experimental procedures, such as the generation of nuclear encoded apicoplast-specific genetic knockouts, which is likely why none have been published since the discovery of the IPP chemical bypass in 2011. Additionally, metabolic labeling experiments are not currently possible due the absence of commercially available isotopically labeled IPP.

In order to facilitate fundamental research into apicoplast biology, we genetically engineered *Plasmodium falciparum* parasites to express enzymes in the cytosol that belong to an alternate route for isoprenoid synthesis called the mevalonate (MVA) pathway. The MVA pathway, found in humans and many other eukaryotes, as well as in archaea and some bacteria, relies on a completely different set of enzymes and intermediates to generate isoprenoids, distinct from the endogenous MEP pathway in *Plasmodium* [11,26]. This line, hereafter referred to as PfMev, converts exogenously supplied mevalonate into isoprenoid precursors in the cytosol of the parasite, and, to our knowledge, represents the first time that a heterologous metabolic pathway has been successfully introduced into *Plasmodium* parasites.

Through the work outlined here, we validated and characterized the PfMev line, showing that mevalonate supplementation provides an effective and affordable tool to probe apicoplast biology. Mevalonate is an inexpensive (about 300 times cheaper than IPP) and stable compound that can support the indefinite culture of PfMev parasites after loss of the apicoplast. We tracked the incorporation of labeled carbons derived from mevalonate into isoprenoid products, demonstrating the functionality of the mevalonate pathway while also providing an example of how downstream isoprenoid products can be tracked and identified. Additionally, we showed that the MVA pathway, in combination with mevalonate, can be employed as a selectable marker to confer parasite resistance against apicoplast-targeting drugs, such as fosmidomycin, thereby expanding the small list of markers that are currently used to generate transgenic parasites. To facilitate downstream genetic modification of the PfMev line, we employed a novel plasmid replicon to prevent unwanted recombination of transfection vectors with the PfMev backbone. As a proof of concept, we were also able to delete the enzyme known to be the target of the drug fosmidomycin, establishing that essential genes can be knocked out in PfMev parasites as long as mevalonate is supplemented. Finally, we explored the consequences of apicoplast disruption on parasite metabolism and gene expression. We examined the transcriptome and metabolome of PfMev parasites treated with azithromycin to disrupt the apicoplast and characterized changes in parasite metabolism and transcription over the course of the intraerythrocytic developmental cycle (IDC). These results provide new insights into apicoplast biology and demonstrate the utility of the PfMev line as a platform for the investigation of essential apicoplast processes.

## Results

### Design and validation of a mevalonate-dependent isoprenoid synthesis pathway

We engineered a genetic bypass system to generate the isoprenoid precursors IPP and DMAPP using enzymes from the mevalonate (MVA) pathway, which are distinct from the enzymes that comprise the endogenous MEP pathway within the parasite apicoplast. Mevalonate pathway

enzymes were chosen from different organisms based on a number of factors including protein length, structure, and prior enzymatic characterization. We designed the bypass system to express three enzymes, which together are capable of converting mevalonate into IPP. We included a fourth enzyme, the *E. coli* IPP isomerase (Idi) that catalyzes the interconversion of IPP and DMAPP. The expression of MVA pathway enzymes in *E. coli* is toxic unless this enzyme is overexpressed, presumably due to improper balancing of IPP/DMAPP levels [27]. Although *P. falciparum* has a putative Idi [28], we expressed the *E. coli* Idi to ensure proper balancing of the increased IPP/DMAPP levels in the cytosol that should occur due to the presence of the MVA pathway (we did not test constructs that lacked *E. coli* Idi).

Synthetic genes encoding all four proteins were optimized for expression in *E. coli* and manually curated to avoid unwanted endonuclease sites and rare codons that might interfere with protein expression in *P. falciparum*. The four genes were separated by sequences that contained a linker region, stop codon and ribosomal binding site (RBS) for polycistronic expression in *E. coli* (**Fig 1A**). *E. coli*, like *Plasmodium*, contains an endogenous MEP pathway that is sensitive to fosmidomycin. Expression of the MVA enzymes in *E. coli* rescued fosmidomycin growth inhibition in a mevalonate-dependent manner, demonstrating the functionality of the pathway in *E. coli* (**Fig 1B**). The stop codon and RBS in each linker were flanked by endonuclease sites, allowing us to remove these bases and fuse adjacent proteins in the four-enzyme pathway. We tested the eight possible combinations of fusion proteins for their ability to function in *E. coli* and found that all combinations bypassed fosmidomycin treatment (**Fig 1B**). These results show that the MVA pathway enzymes are functional in *E. coli* and that they can be fused into a multifunctional polypeptide.

## Generation and validation of the PfMev parasite line

We next attempted to express the MVA pathway enzymes in *P. falciparum*. All parasite transfection vectors currently use the pUC replicon. However, we wanted to limit the homology between the MVA plasmid and any plasmids used for subsequent genetic modification, and therefore chose the replicon from pACYC, which contains a p15a origin and chloramphenicol resistance cassette [29]. Although plasmids containing a p15a origin have a low copy number in *E. coli* (5 per cell), we were able to increase the copy number 5-fold by introducing a point mutation in the structural RNA (RNA I) that is thought to maintain plasmid copy number and compatibility group [30]. The new origin is referred to as p15HC (high copy) in **S1A Fig**. We also reduced the size of the replicon by removing regions proposed to be important for terminating the origin template RNA [31]. The resulting replicon is small (1.3kbp) and allowed us to add the MVA pathway genes, as well as an *att*P site to facilitate integration using Bxb1 mycobacteriophage integrase [32]. We transfected this plasmid (**S1B Fig**) into NF54$^{attB}$ parasites [33] and selected for integrants with fosmidomycin in the presence of 50μM mevalonate (supplemented as racemic mevalonolactone), generating the PfMev parasite line (**Fig 2A**).

We next investigated how much mevalonate is required to bypass fosmidomycin inhibition. We treated parasites with 25μM fosmidomycin (50x IC$_{50}$), and supplemented the parasites with various concentrations of mevalonate. As shown in **Fig 2B**, mevalonate concentrations of 10μM or more successfully bypassed the lethal effect of fosmidomycin [17]. These data also indicate that the engineered mevalonate pathway is sufficient for the generation of essential isoprenoid precursors to support parasite growth.

## Apicoplast labeling for studying organelle morphology

In addition to the MVA pathway enzymes, the PfMev parasite line also expresses the Super Folder Green (SFG) protein [34], with the apicoplast leader sequence of *P. falciparum* acyl-

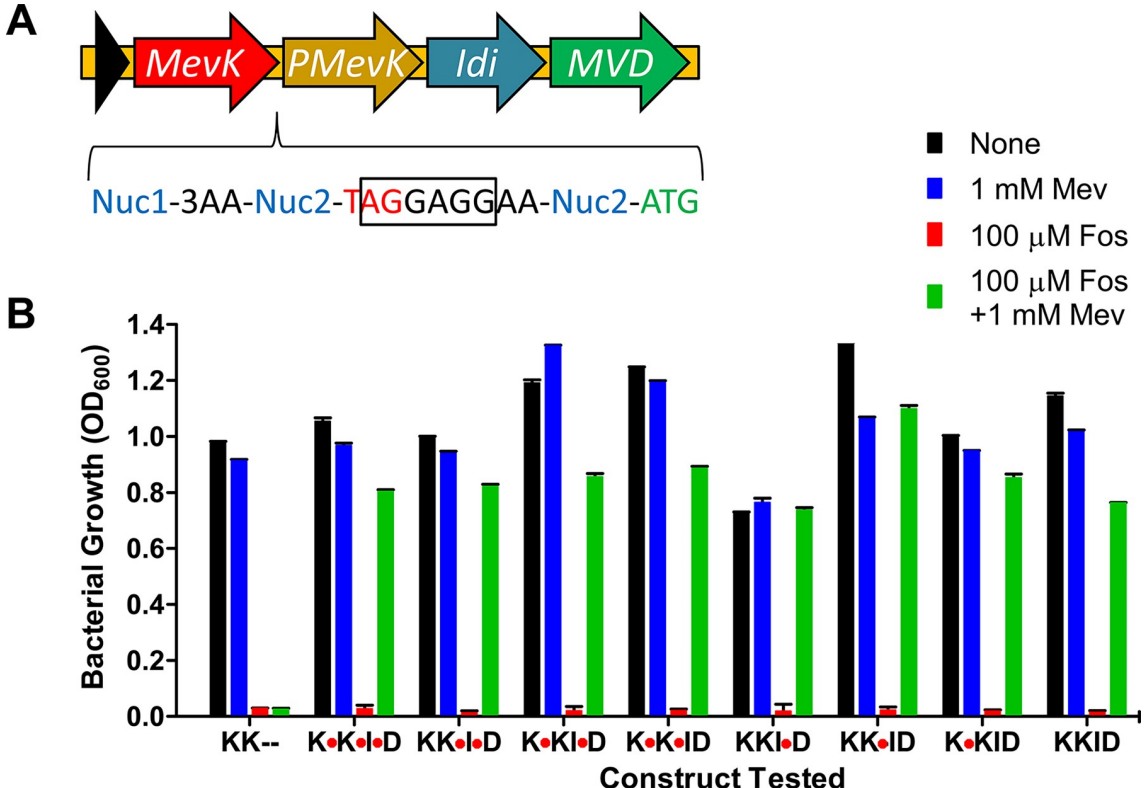

**Fig 1. Design and validation of a multi-protein expression system for mevalonate pathway genes. A)** *Streptococcus pneumoniae* mevalonate kinase (*mevK*), *Homo sapiens* phosphomevalonate kinase (*pmevK*), *E. coli* isopentenyl diphosphate isomerase (*idi*), and *S. pneumoniae* mevalonate 5-diphosphate decarboxylase (*mvd*) were arranged for polycistronic expression in *E. coli* with a stop codon (red) and ribosomal binding site (black box) between each pair of genes. Digestion with a specific endonuclease (Nuc2) allowed these elements to be removed for expression of a multi-functional proteins linked by flexible three amino acid linkers (3AA). **B)** The eight possible ways of connecting the four genes (red octagons show retention of the stop codon) were tested by attempting to rescue the growth of *E. coli* inhibited with the isoprenoid synthesis inhibitor fosmidomycin (Fos). Successful growth in the presence of mevalonate (Mev) showed that all eight constructs rescued fosmidomycin inhibition in a mevalonate-dependent manner, while the negative control only containing the two kinases (KK—) did not.

carrier protein (ACP) appended to its N-terminus. This results in SFG being trafficked to the apicoplast and fluorescently labeling the organelle (api-SFG) [4]. Previous work has demonstrated that under IPP supplementation, certain drug treatments or genetic perturbations can result in disruption of the apicoplast, and the fluorescent labeling of the organelle allows for a phenotypic readout [14,17]. Using live epifluorescence microscopy, we observed that SFG is trafficked to a subcellular compartment consistent with the morphology of the apicoplast organelle (**Fig 3A**). Trafficking of api-SFG to the apicoplast was confirmed using immunofluorescence (IFA) by co-localizing api-SFG with ACP–a known marker of the apicolast organelle [4] (**Fig 3B**).

## Mevalonate rescue of apicoplast-disrupted PfMev parasites

Fosmidomycin treatment, while lethal to *P. falciparum* parasites, does not affect apicoplast integrity [17]. We asked whether mevalonate supplementation could also rescue parasites from antibiotics that disrupt apicoplast maintenance. To accomplish this, we generated apicoplast-disrupted parasites by treating PfMev parasites with 100nM azithromycin (1x IC$_{50}$) supplemented with 50μM mevalonate for one week. Post-treatment, these parasites were

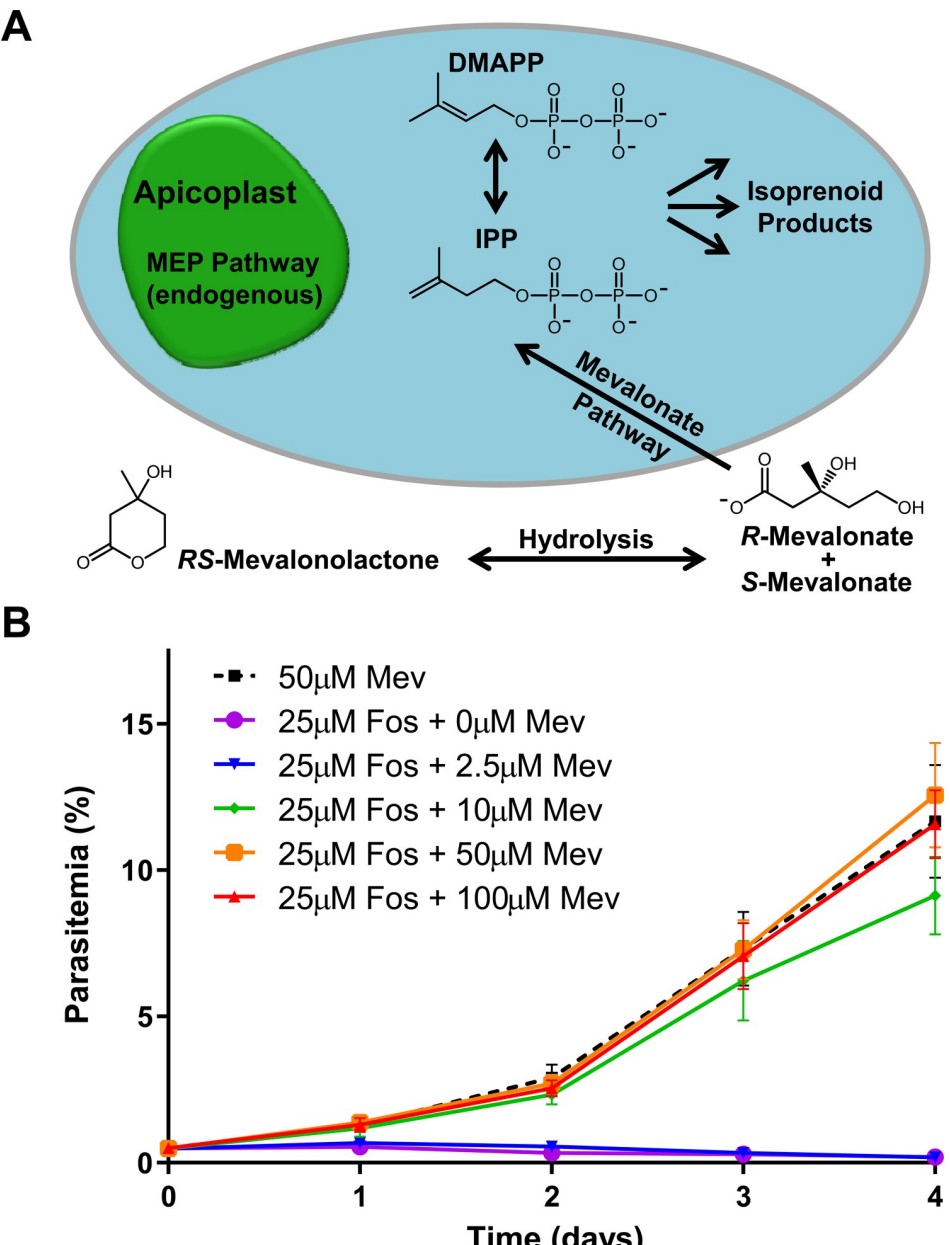

**Fig 2. The MVA pathway bypasses fosmidomycin toxicity in PfMev parasites. A)** Diagram of the parasite cell showing that the endogenous MEP pathway is localized with the apicoplast organelle of the parasite, with IPP and DMAPP presumably being exported out of the organelle. We engineered the PfMev parasites to express four enzymes forming a MVA pathway in the cytosol capable of producing IPP and DMAPP from exogenous mevalonate supplied in the growth medium. **B)** PfMev parasites were treated with 25μM fosmidomycin (50x IC50), and supplemented with various concentrations of mevalonate, with growth compared to an untreated control. PfMev parasites grew to wild-type levels in the presence of a typically lethal concentration of fosmidomycin when supplemented with at least 10μM mevalonate, indicating usage of the engineered mevalonate pathway for the generation of essential isoprenoid precursors. PfMev parasites treated with fosmidomycin and not supplemented with mevalonate failed to grow. These data are from quadruplicate independent experiments, each conducted in quadruplicate with error bars representing the standard error of the mean (SEM).

permanently dependent on mevalonate for survival, indicating that azithromycin disrupted the apicoplast, as previously described [17]. Consistent with this result, we failed to obtain

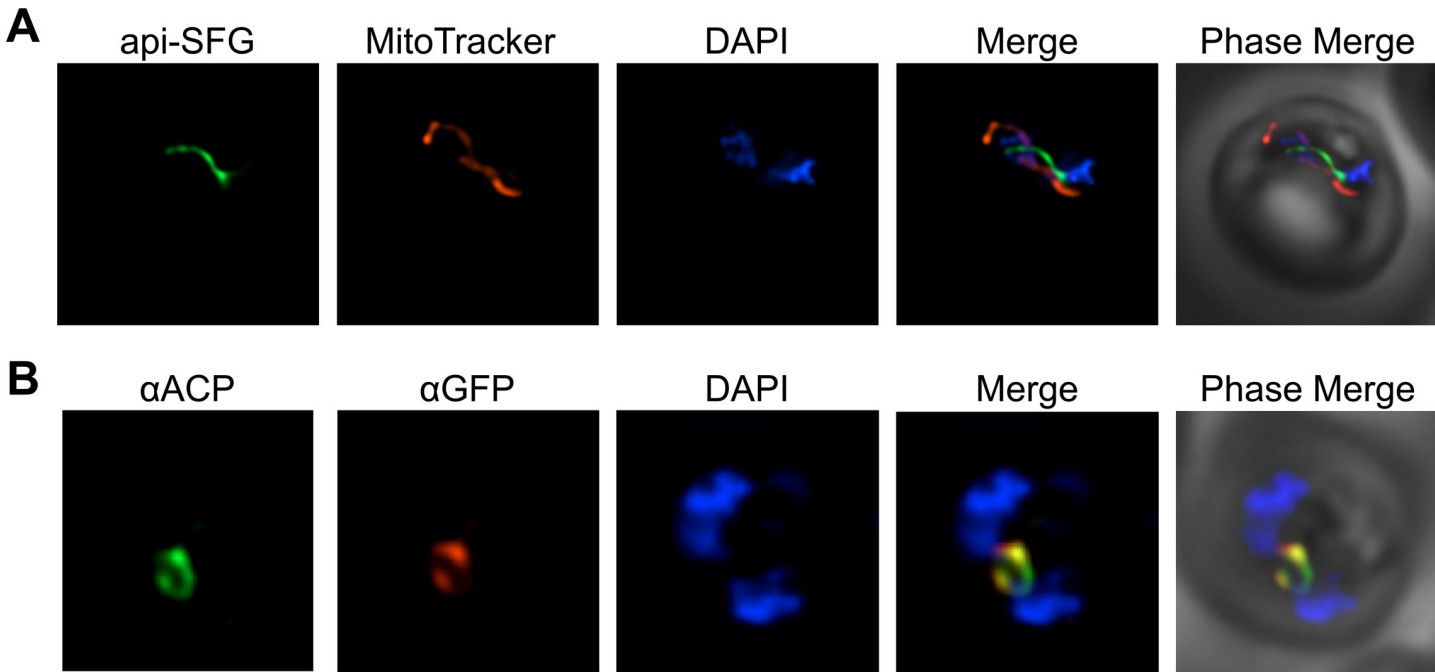

**Fig 3. Fluorescent labeling of the apicoplast organelle in PfMev parasites and confirmation via co-localization. A)** Live fluorescence microscopy of the PfMev line expressing the signal sequence and transit peptide from ACP fused to SFG (green), the mitochondria was stained with MitoTracker (red), and nuclear DNA was stained with DAPI (blue). **B)** Immunofluorescence co-localization of aSFG with the apicoplast marker ACP, with α-ACP (green), α-GFP (red), with nuclear DNA stained with DAPI (blue) in fixed PfMev parasites. Images depict fields that are 10 microns long by 10 microns wide.

PCR amplification products from the apicoplast organellar genome [17] (**Fig 4A**). We examined parasites by live epifluorescence microscopy and were unable to locate intact apicoplasts, but instead observed the presence of multiple discrete api-SFG labeled vesicles, indicative of organelle loss [14] (**Fig 4B**). IFA co-localization experiments showed that ACP is also found in these api-SFG vesicles providing further evidence that the apicoplast organelle is disrupted (**Fig 4C**). To determine the minimum concentration of mevalonate required to rescue apicoplast-disrupted parasites, we compared the growth of these parasites under various concentrations of mevalonate to that of apicoplast-intact PfMev parasites. We found that supplementation with 10µM mevalonate, or more, restored the growth of apicoplast-disrupted parasites to near wild-type levels (**Fig 4D**).

## Metabolic labeling of PfMev parasites with [2-$^{13}$C]-mevalonate

We conducted metabolic labeling of PfMev parasites to confirm that exogenously supplemented mevalonate was being used by the MVA enzymes introduced into the parasite to form IPP and downstream isoprenoids. We generated parallel PfMev parasite cultures, both treated with 25µM fosmidomycin (50x IC$_{50}$), one supplemented with 50µM unlabeled mevalonate (**Fig 5A**) and the other with [2-$^{13}$C]-mevalonate (**Fig 5B**). After 48 hours of growth, metabolites were extracted and analyzed via targeted liquid-chromatography mass spectrometry (LC-MS/MS) using selected reaction monitoring (SRM) [35]. Using this method, we were able to detect the isoprenoid precursor IPP (and/or the isobaric isomer DMAPP) (**Fig 5A and 5B**) in addition to the downstream isoprenoid product farnesyl pyrophosphate (FPP) (**Fig 5C and 5D**). We observed clear mass shifts of these metabolites between unlabeled and labeled parasite samples, demonstrating incorporation of [$^{13}$C] carbon in the labeled samples [36].

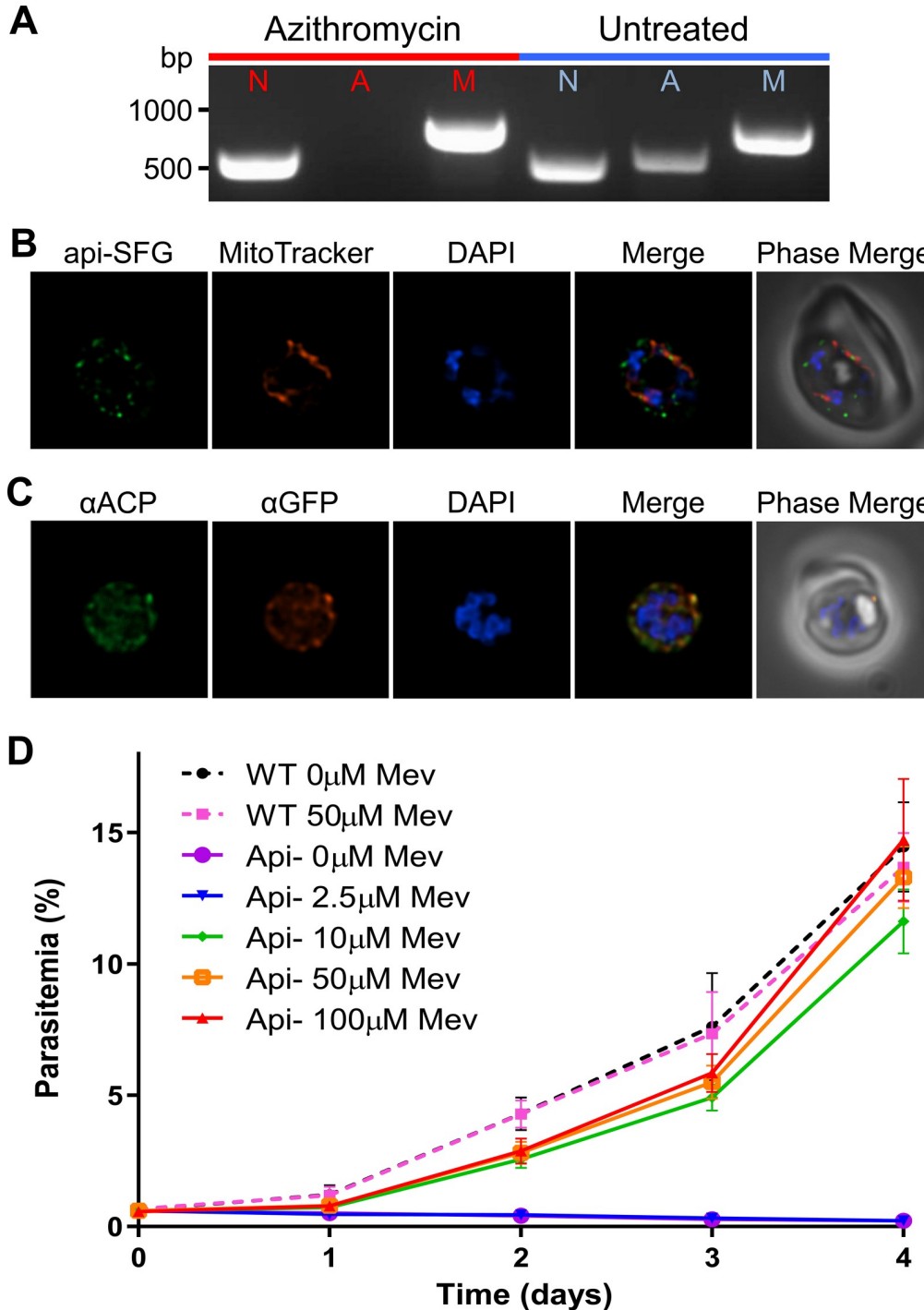

**Fig 4. Generation and characterization of an apicoplast-disrupted PfMev parasite line. A)** Attempted detection of nuclear, mitochondrial, and apicoplast genomes by PCR amplification of the genes *ldh* (nuclear, N), *sufB* (apicoplast, A), and *cox1* (mitochondrial, M) from PfMev parasites treated with 100nM azithromycin and 50μM mevalonate for one week (red), in addition to an untreated control (blue). Failure to amplify *sufB* in PfMev parasites treated with azithromycin/mevalonate indicates loss of the apicoplast organelle. **B)** Live fluorescence microscopy of PfMev parasites after treatment with 100nM azithromycin and 50μM mevalonate for one week, showing a disrupted organelle phenotype, with multiple discrete vesicles instead of a single intact organelle. The apicoplast is labeled with the api-SFG protein (green), the mitochondrion is stained with MitoTracker (red), and nuclear DNA is stained with DAPI (blue). **C)** Immunofluorescence co-localization of api-SFG with the apicoplast marker ACP, with αGFP (red), αACP (green), with nuclear DNA stained with DAPI (blue) in fixed apicoplast disrupted PfMev parasites. **D)** Growth curve of

apicoplast-disrupted PfMev parasites grown in the presence of different concentrations of mevalonate. The growth of untreated (apicoplast intact) PfMev parasites was also measured in the presence of 50μM or 0μM mevalonate as a control. Supplementation with 10μM mevalonate, or more, restored the growth of apicoplast-disrupted parasites to near wild-type levels. Data from quadruplicate independent experiments, each conducted in quadruplicate are shown with error bars representing the standard error of the mean (SEM). Images depict fields that are 10 microns long by 10 microns wide.

Furthermore, we were able to show that the endogenous MEP pathway was inhibited by fosmidomycin, since this pathway would produce unlabeled products independent of mevalonate labeling. Taken together, these data clearly demonstrate usage of mevalonate by the MVA pathway enzymes to generate the isoprenoid precursors IPP/DMAPP and downstream isoprenoids such as FPP.

## Deletion of an apicoplast drug target using Cas9-mediated genome editing

Supplementation of PfMev parasites with mevalonate enables us to disrupt the apicoplast, while keeping parasites viable, and thus should also allow for the deletion of essential apicoplast-specific genes. In order to test this, we chose to delete the target of fosmidomycin, DXPR (PF3D7_1467300; 1-deoxy-*D*-xylulose 5-phosphate reductoisomerase) [37]. Δ*dxpr* parasites were obtained using Cas9-mediated editing [38] under continuous supplementation with 50μM mevalonate, with deletion of DXPR confirmed via PCR (**Fig 6A** **and S2 Fig**). The phenotype of the apicoplast was then ascertained in order to determine if DXPR deletion had any effect on the organelle. We were able to detect a gene from the apicoplast genome (*sufB*) in the Δ*dxpr* parasites (**Fig 6B**), and we observed a single intact apicoplast using live epifluorescence microscopy (**Fig 6C**). However, Δ*dxpr* parasites quickly died upon removal of mevalonate from culture media (**Fig 6D**). Collectively, these results demonstrate that DXPR is essential for parasite survival, but also show that the activity of DXPR is not needed for maintaining the genome or morphology of the apicoplast organelle. Overall, our experiments show that we can delete an essential apicoplast drug target using the mevalonate bypass system, allowing us to explore the essential biology of the apicoplast organelle.

## Transcriptomic analysis of apicoplast-disrupted parasites

Treatment with antibacterial drugs, such as azithromycin, leads to disruption of the apicoplast and subsequent death unless the parasites are rescued with isoprenoid precursors. We used mevalonate supplementation in the PfMev line to study the metabolic consequences of organelle disruption after a period of azithromycin treatment. Synchronized ring-stage parasites were cultured for 48 hours with samples taken every eight hours for both transcriptomic and metabolomic analysis. Expression of genes from the apicoplast genome were not detected at significant levels in apicoplast-disrupted parasites, consistent with loss of the organellar genome (**Fig 7A**). We correlated global gene expression from ring, trophozoite, and schizont stage apicoplast disrupted parasites with corresponding data from each time point generated in the previously published high-resolution 3D7 IDC transcriptome [39]. Global gene expression from the apicoplast-disrupted parasites correlated closely with similar time points from the high-resolution data set (**Fig 7B**). A control data set collected from the parental parasites used to generate the PfMev line behaved similarly (**Fig 7B**). The close correlation of the control and apicoplast-disrupted data with the high-resolution data set demonstrates that there is no significant perturbation to the 48-hour developmental cycle resulting from disruption of the apicoplast. We next addressed the question of whether nuclear-encoded apicoplast proteins, such as those that comprise the MEP pathway, are differentially expressed in apicoplast-

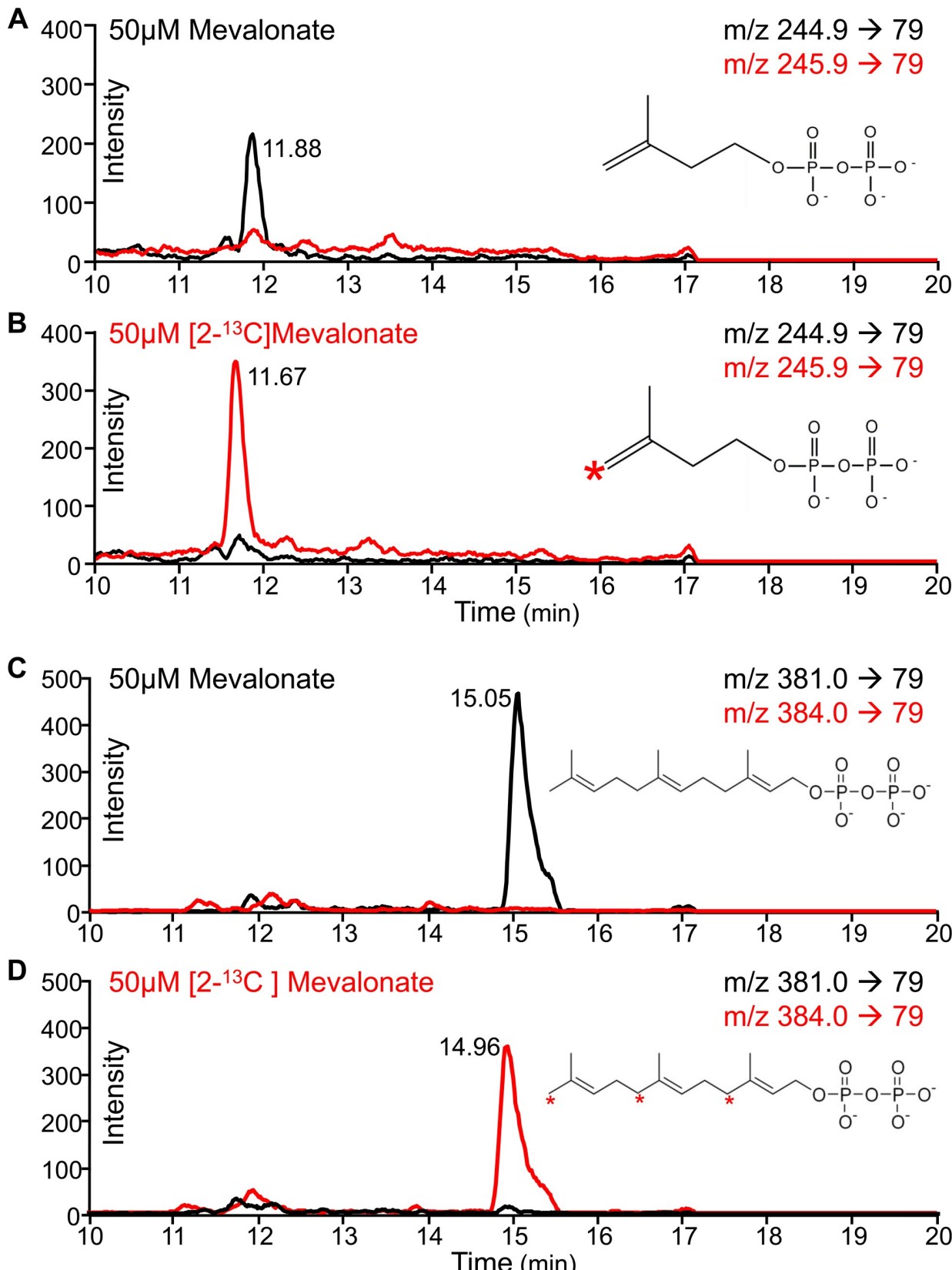

**Fig 5. Metabolic labeling of PfMev parasites with [2-¹³C]-mevalonate.** PfMev parasite cultures were treated with 25μM fosmidomycin (50x IC50) and supplemented with either 50μM unlabeled mevalonate or [2-¹³C]-mevalonate for 48 hours. Metabolites were extracted and analyzed via targeted liquid-chromatography mass spectrometry (LC-MS/MS) using selected reaction monitoring (SRM). **A)** Detection of IPP/DMAPP (black) in parasites treated with 50μM unlabeled mevalonate and 25μM fosmidomycin, with attempted detection of mass-shifted IPP/DMAPP (red). **B)** Detection of mass-shifted IPP/DMAPP (red) in parasites treated with 50μM [2-¹³C]-mevalonate and 25μM fosmidomycin, with attempted detection of IPP/DMAPP (black). **C)** Detection of FPP (black) in parasites treated with 50μM mevalonate and 25μM fosmidomycin, with attempted detection of mass-shifted FPP (red). **D)** Detection of mass-shifted FPP (red) in parasites treated with 50μM [2-¹³C]-mevalonate and 25μM fosmidomycin, with attempted detection of FPP (black). The red asterisks on the biomolecules shown denote the positions of the labeled carbons. Metabolite peaks are labeled with the liquid chromatography column retention time in minutes.

disrupted parasites compared to control parasites. We analyzed a list of 96 high-confidence apicoplast genes [10] and found that there was a similar pattern of gene expression in the control parasites compared to the apicoplast-disrupted parasites (**Fig 7C and S3 Fig**). Overall, transcriptomic analysis of apicoplast-disrupted PfMev parasites indicates that cell cycle progression and the expression of genes from the nuclear genome are largely unaffected by loss of the apicoplast organelle. Surprisingly, expression of apicoplast proteins encoded in the nuclear genome continues despite loss of their organellar destination, highlighting a lack of communication between the parasite cell and its endosymbiont organelle.

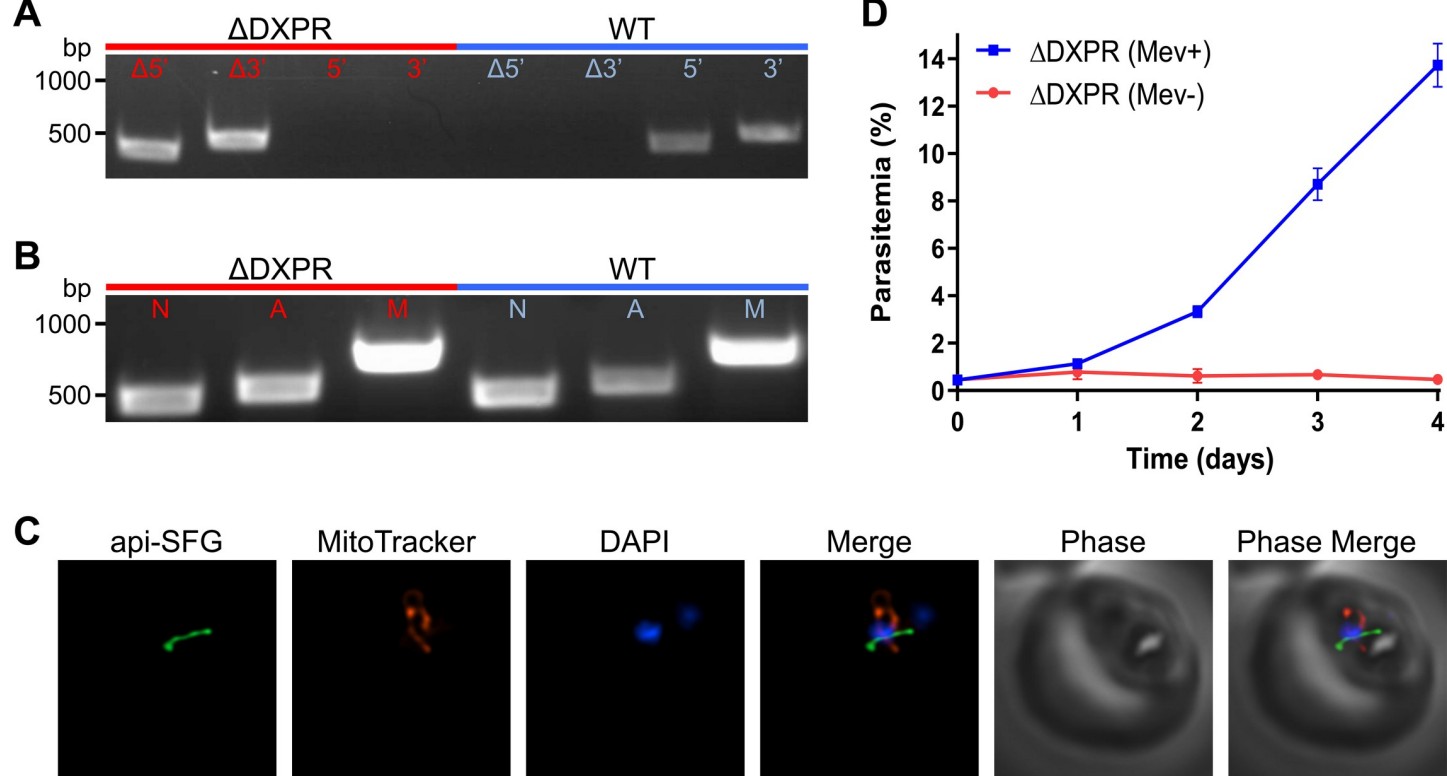

**Fig 6. Deletion of the fosmidomycin drug target DXPR. A)** Confirmation of DXPR gene deletion via PCR showing the expected 5' and 3' integration of the selection cassette. Δ*dxpr* parasites (red) were also screened for the unmodified 5' and 3' loci to detect if there were any residual wild-type parasites and compared to a control (blue). **B)** PfMev Δ*dxpr* parasites were screened for the presence of the nuclear (*ldh*), apicoplast (*sufB*), and mitochondrial (*cox1*) genomes via PCR (red), and compared to a control (blue). Both lines showed the presence of nuclear, apicoplast and mitochondrial genomes, indicating deletion of DXPR does not result in organelle loss. **C)** Live epifluorescence microscopy of the PfMev Δ*dxpr* parasites showing the presence of an intact organelle. The apicoplast is labeled with the api-SFG protein (green), the mitochondrion is stained with MitoTracker (red), and nuclear DNA is stained with DAPI (blue). Images depict fields that are 10 microns long by 10 microns wide. **D)** PfMev Δ*dxpr* parasites grown in the presence of 50μM mevalonate (blue) or no mevalonate (red). Error bars represent the standard error of the mean from three independent experiments, each conducted in quadruplicate. The reliance of PfMev Δ*dxpr* parasites on mevalonate for growth indicates that DXPR is essential for the blood-stage survival of the parasite.

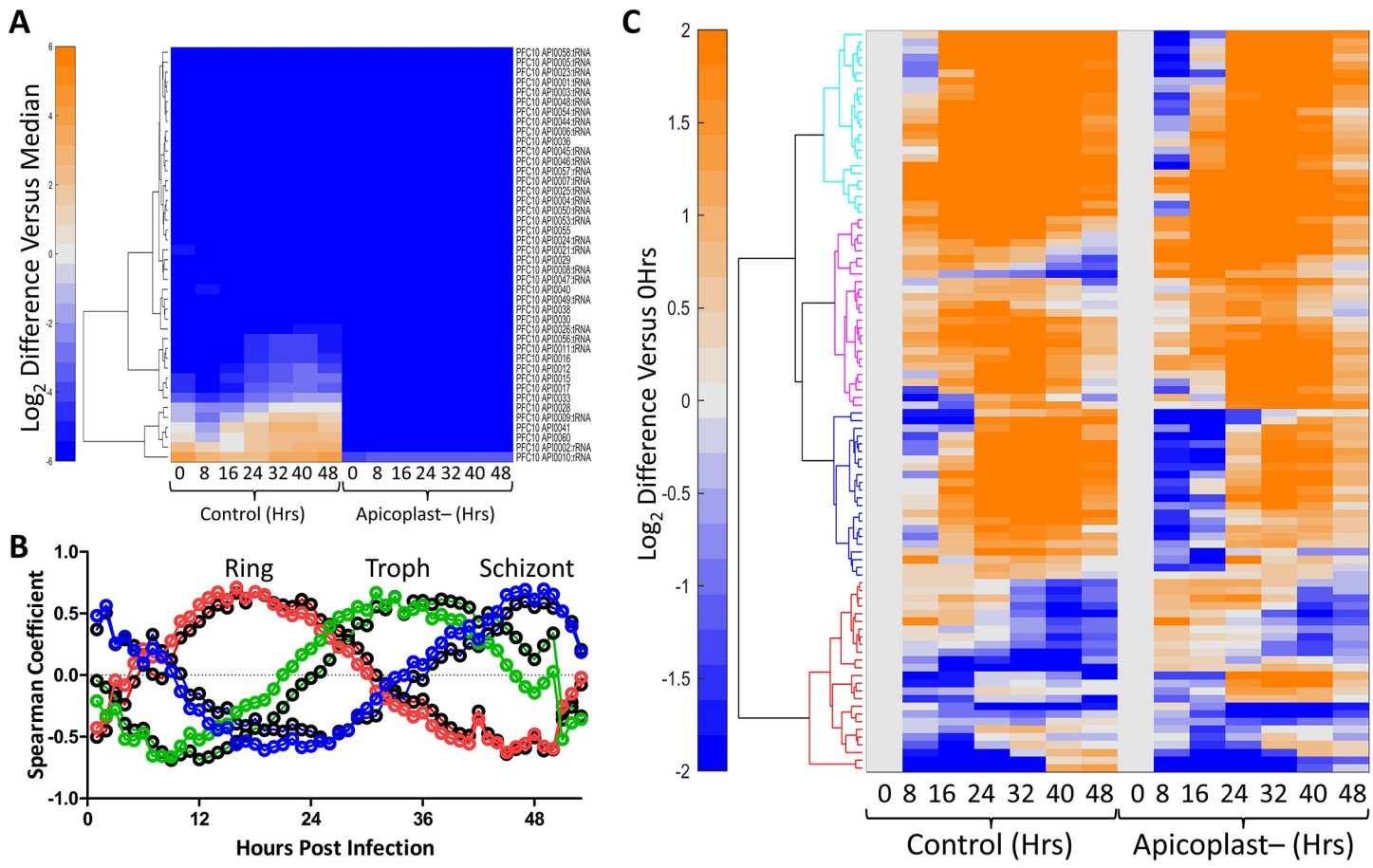

**Fig 7. Transcriptomic analysis of apicoplast-disrupted parasites. A)** Hierarchical clustering of apicoplast genome expression from control and apicoplast-disrupted parasites during the IDC. The apicoplast gene expression data were normalized by the median expression of 5851 genes from corresponding parasites at each time point before performing the clustering. We used Euclidean distance measure, along with Ward's method, to create the hierarchical clusters in MATLAB. The expression profile of the apicoplast genes from the apicoplast-disrupted parasites is minimal confirming that the organelle is missing. The values are shown on a $\log_2$ scale. **B)** Spearman coefficient values were calculated by comparing the global transcriptional data generated from control (black circles) and apicoplast disrupted (colored circles) parasites at the indicated time points against corresponding data from each time point generated in the high-resolution 3D7 IDC transcriptome from Llinás and coworkers [39]. The y-axis shows Spearman coefficient; the x-axis shows hours post invasion in the IDC data set. The apex of the peak in each graph corresponds to the approximate point in the IDC to which apicoplast-disrupted parasites best correlate at the indicated incubation time. Plots are shown with a Loess fit of the data: 16hrs (red), 32hrs (green), and 48hrs (blue). **C)** Hierarchical clustering of transcriptomic data, of 96 high-confidence apicoplast genes, from control and apicoplast-disrupted parasites at the indicated time points. The expression profile of each gene was normalized by its expression at time point 0 to visualize variation in their temporal profile, in response to apicoplast disruption. Four distinct clusters are shown (cyan, magenta, blue, and red) and are further described in **S3 Fig**. The clustering method used here is identical to that described for A).

## Metabolomic analysis of apicoplast-disrupted parasites

We tracked the relative abundance of metabolites in parallel cultures of infected and uninfected RBCs following the same time course as above. At each time point of the IDC, we compared the metabolite profiles to determine which metabolic changes were due to parasite metabolism as opposed to RBC metabolism. We conducted this metabolic analysis for apicoplast-disrupted parasites as well as control parasites to identify metabolic perturbations that accompany loss of the organelle. **Fig 8A** shows temporal changes in 405 metabolites, which are present in both control and apicoplast disrupted parasites, over the course of parasite development. Loss of the apicoplast organelle does not seem to affect the levels of most metabolites over the course of the intraerythrocytic developmental cycle. However, hierarchical clustering of these results reveals four clusters of metabolites which do differ in apicoplast-disrupted

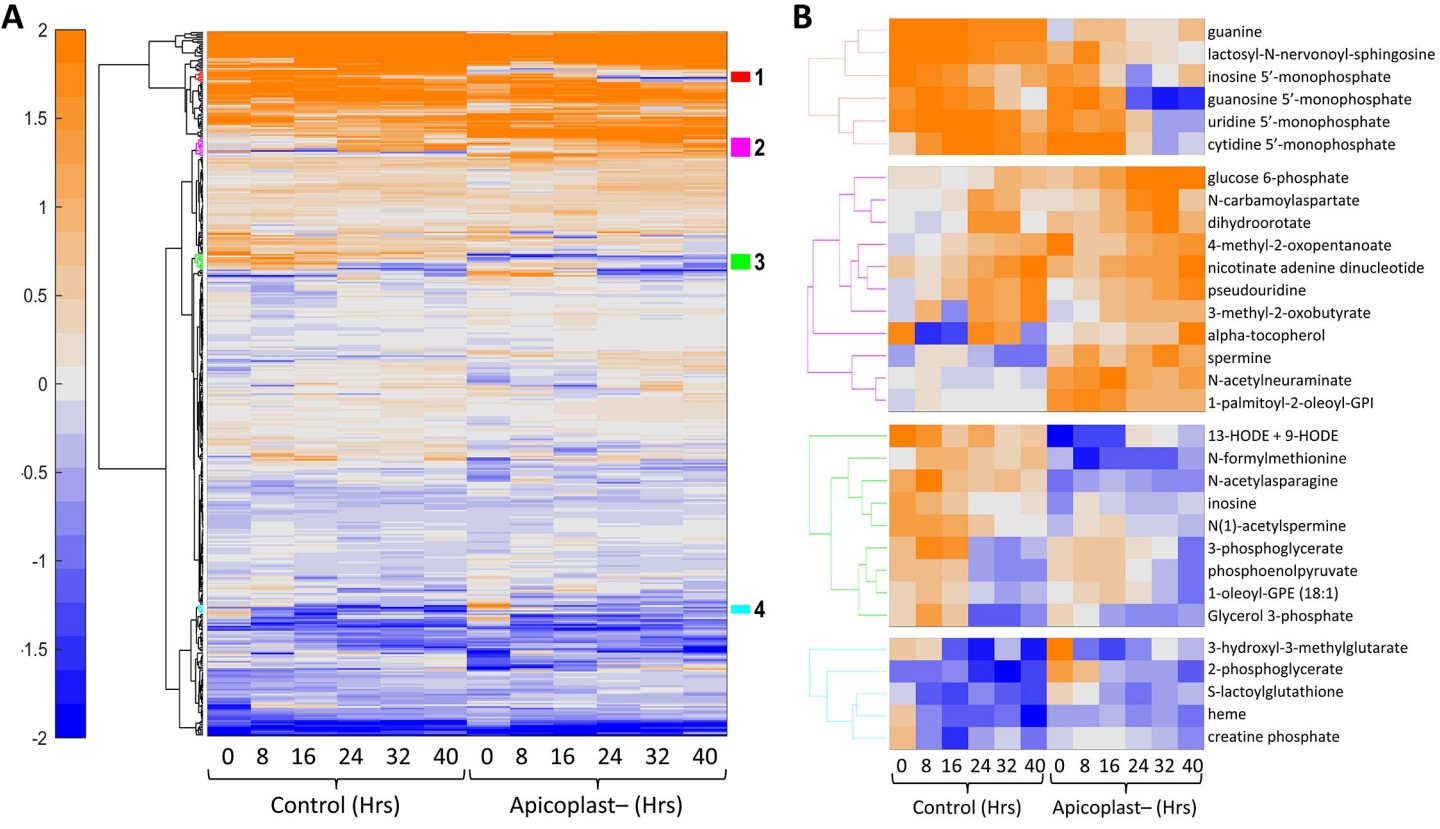

**Fig 8. Metabolomic analysis of apicoplast-disrupted parasites. A)** Hierarchical clustering of metabolomic data from control and apicoplast-disrupted parasites at the indicated time points. Metabolomic data are shown for 405 metabolites that are present in both data sets. The data represent the abundance of each metabolite after normalization with their median value at the sampled time points. The hierarchical clustering method is the same as described above, except that we also used the k-nearest-neighbor method to impute any missing values in the data. The values are shown on a log$_2$ scale with orange and blue colors indicating amounts higher and lower than the median value, respectively. **B)** Four clusters of metabolites that have temporal profiles which differ between control and apicoplast-disrupted parasites are colored in red, magenta, green, and cyan.

parasites (**Fig 8B**). Some of these metabolites may be normally produced by the apicoplast, and therefore are more abundant in the control parasites. Other metabolites appear to accumulate in apicoplast-disrupted parasites indicating some compensation or perturbation associated with loss of the organelle (**Fig 8B**). In particular, there are changes in nucleotide metabolites as well as some central carbon and lipid metabolites.

## Discussion

The MVA pathway is a popular tool used in synthetic biology, particularly for the engineering of biofuels [40,41] and for the production of terpenoids for use in the chemical synthesis of complex compounds, including the antimalarial compound artemisinin [27]. *E. coli* has proven to be a workhorse for isoprenoid production and often the six-enzyme yeast MVA pathway is used to produce high levels of IPP for further modification [42]. In these MVA systems, the *E. coli* IPP isomerase typically has to be overexpressed to balance the IPP/DMAPP ratio [27] and avoid IPP toxicity [43]. We chose to express the *E. coli* IPP isomerase in conjunction with the last three enzymes of the MVA pathway. Unlike the complete MVA pathway that uses the common metabolite acetyl-CoA, our MVA pathway relies on mevalonate, a cheap, stable, non-toxic, cell-permeant metabolite that is not present in malaria parasites or parasite growth medium. This allows us to conditionally control the activity of the MVA

pathway through mevalonate supplementation. Rather than use the yeast MVA enzymes, we chose enzymes from different sources based on several properties including size, structural information, activity (including feedback inhibition), multimer state, and the demonstrated ability to add protein or peptide tags to the termini of the proteins. By screening for activity in *E. coli*, we found that these enzymes are functional when linked together as a single polypeptide (**Fig 1**). Although multi-functional MVA proteins do not seem to exist in nature, there may be an advantage to physically connecting the enzyme domains since this arrangement would tend to increase the local concentration of pathway intermediates.

There are several relevant forms of mevalonate available commercially. Using mevalonic acid phosphate as a substrate would potentially reduce the complexity of the engineered mevalonate pathway since the mevalonate kinase enzyme would not be needed. Similarly, using mevalonic acid pyrophosphate would bypass both of the kinases in the MVA pathway. Unfortunately, these phosphorylated compounds are prohibitively expensive. The biologically active stereoisomer *R*-mevalonic acid is also surprisingly expensive and is listed in the Sigma catalog for about $18,000 per gram. By contrast, the ring or lactone form of mevalonate (mevalonolactone) is significantly cheaper. *R*-mevalonolactone costs ten times less than *R*-mevalonic acid and the racemic form costs only $163 per gram (enough to make 150L of growth medium at 50μM). We focused our validation efforts on racemic mevalonolactone due to the affordability of this compound and also because $^{13}C$ heavy atom labeled forms of mevalonate are only available as racemic mevalonolactone. As shown in **Fig 2B** and **Fig 4D**, 10μM of racemic mevalonolactone is enough to support parasite growth, whereas 2.5μM is not enough. Presumably, 10μM of racemic compound corresponds to 5μM of the biologically active *R* stereoisomer. This concentration is significantly lower than the 200μM concentration of IPP needed to support parasite growth [14]. It is likely that lower concentrations of IPP could support parasite growth, but problems with the uptake and stability of IPP, which has a half-life of 4.5 hours at 37°C [25], probably limit the amount available for isoprenoid synthesis.

The apicoplast MEP pathway generates isoprenoid precursors, which are used to form a class of downstream biomolecules called isoprenoids that are essential for parasite survival [44]. To date, isoprenoids such as ubiquinone [45], menaquinone [46], dolichols [47], protein prenyl tails [48], and carotenoids [49] have been found in *P. falciparum*. More recently, an entire subclass of isoprenoids called monoterpenes, typically found in plants, were discovered and are believed to function as volatile chemoattractants for mosquitoes [50]. While the MEP pathway has received attention as an attractive source of potential drug targets [51,52], the identity and essentiality of the full complement of isoprenoids generated by the parasite remains to be fully elucidated. Isoprenoids are the most abundant class of small molecules [53,54] and further discovery of these compounds is likely to provide additional insights into parasite biochemistry and may allow for the identification of additional druggable pathways. Through our metabolic labeling experiments we were able to track the incorporation of [2-$^{13}C$]-mevalonate into downstream products such as IPP and FPP. While our motivation was to validate the function of the engineered MVA pathway, these results provided a proof of concept demonstrating the utility of this line for the possible detection and identification of parasite-generated isoprenoids. Previous attempts to identify isoprenoids in *P. falciparum* relied on the incorporation of trace amounts of expensive radioactive compounds such as [$^{3}H$] FPP, [$^{3}H$]GPP, and [$^{3}H$]GGPP [45,47,49,55,56]. As shown in **Fig 5**, using the PfMev parasites we can completely inhibit the endogenous MEP pathway, thereby forcing the parasites to be completely reliant on the labeled mevalonate for isoprenoid generation, resulting in very efficient and uniform labeling of downstream products.

In organisms such as yeast, the mevalonate pathway contains seven enzymes. The first three enzymes comprise the 'upper mevalonate pathway' which can convert acetyl-CoA into

mevalonate, while the 'lower mevalonate pathway' converts mevalonate into the isoprenoid building blocks [26]. We introduced the lower mevalonate pathway in PfMev parasites in order to make the pathway controllable through supplementation with mevalonate, a metabolite not produced by the parasite. This system could potentially be introduced into *P. berghei* for bypass of the apicoplast in an *in vivo* model. However, it may be technically challenging to increase the serum concentration of mevalonate from physiological levels of about 200nM to the ~10μM likely required for bypass [57]. Alternatively, the additional three enzymes from the upper portion of the pathway could be added to complete the mevalonate pathway. While the complete pathway would likely work for bypass, investigators would not be able to control pathway activity through nutrient supplementation since acetyl-CoA is a basic metabolite in malaria parasites.

Prior attempts to investigate the essentiality of apicoplast-specific proteins have been dependent on conditional knockdown approaches and direct supplementation with IPP [15–19]. Using CRISPR-Cas9 in PfMev parasites, we were able to generate the first targeted deletion of a gene encoding an essential apicoplast-specific protein. Gene deletion, which is a relatively longer-term experiment, often taking 30+ days to generate, was facilitated by certain advantageous features associated with this line, namely the relatively stability and inexpensiveness of mevalonate. The PfMev line was also engineered to express an apicoplast localized green fluorescent protein (SFG) allowing for rapid analysis of the resulting organelle phenotype. Importantly, these mevalonate-dependent knockout parasites can be cryopreserved, a critical requirement for archiving and sharing genetically modified parasite lines. In addition to longer-term experimentation, the features associated with this line should also facilitate larger-scale analysis of genes encoding essential apicoplast-specific proteins, such as forward genetic screens. Overall, we have demonstrated the utility of the PfMev line as a user-friendly platform that easily facilitates the genetic manipulation of essential apicoplast-specific genes, allowing for the investigation of their roles in parasite survival and organelle maintenance.

While previous work has shown that certain nuclear encoded apicoplast-specific proteins continue to be expressed upon apicoplast disruption [14,17,58,59], there have not been any studies looking at how apicoplast disruption affects transcription globally. Upon the analysis of PfMev parasites lacking an apicoplast we were surprised to find that parasite growth, the developmental cycle, and gene expression are largely unaffected by organelle loss (**Fig 7**). Surprisingly, the expression of nuclear-encoded apicoplast proteins follows the same transcriptional program despite the absence of an organellar destination for these proteins. Presumably, these proteins accumulate in the GFP-positive vesicles observed in **Fig 4**. ACP also accumulates in these vesicles (**Fig 4C**), as observed by others [14,17]. Our results suggest that the full complement of nuclear-encoded apicoplast proteins may exist in these vesicles, including all six of the MEP pathway enzymes. Although these enzymes are expressed, they probably fail to generate isoprenoid precursors because they lack necessary conditions (substrates, cofactors, apicoplast-encoded chaperones, etc.) for pathway activity. It is possible, however, that other apicoplast proteins remain functional in the vesicles that accumulate after apicoplast disruption.

Similarities in the global transcriptional program of control and apicoplast disrupted parasites are mirrored in metabolomic data (**Fig 8**). For most of the 405 metabolites observed in both data sets, metabolite levels followed the same pattern of production and consumption despite loss of the organelle. Hierarchical clustering identified four clusters of metabolites that differ between the control and apicoplast disrupted parasites. Some metabolites, such as N-formylmethionine, are thought to be produced in the apicoplast [60] and it makes sense that their levels would decrease after loss of the organelle. The isoprenoid α-tocopherol (vitamin E) was predicted to rely on the apicoplast MEP pathway [56], but was found at higher levels in

apicoplast-disrupted parasites. This is perhaps an indication that 50μM mevalonate supports an increased level of isoprenoid synthesis by the MVA pathway compared to the native MEP pathway. Loss of the apicoplast seems to be associated with reduced levels of nucleoside monophosphates (NMPs) and nucleobases. While organellar nucleotide metabolism no longer occurs in apicoplast-disrupted parasites, there may be a more global effect responsible for these reduced levels. Similarly, increases in some central metabolites and metabolites of lipid and sphingolipid metabolism suggest that there are metabolic shifts that result from or compensate for loss of the apicoplast.

Through the work outlined here, we were successful in engineering *P. falciparum* parasites to express an exogenous MVA isoprenoid precursor pathway, allowing for efficient bypass of the apicoplast organelle through mevalonate supplementation. Characterization of the resulting PfMev line demonstrated its utility as a tool for the investigation of the apicoplast, with applications including isoprenoid labeling and the deletion of genes encoding essential apicoplast specific proteins. Additionally, we collected and analyzed transcriptomic and metabolomic data over the course of the complete IDC for PfMev parasites lacking an apicoplast. These results will lay the groundwork for systems biology approaches, such as metabolic modeling, to understand the impact of organelle loss on parasite metabolism. Furthermore, certain advantageous features associated with this line, such as the ability to easily cryopreserve mevalonate-dependent lines, and the affordability of the bypass compound should allow for longer-term and larger-scale experimentation, such as forward genetic screens, or the screening of drug compound libraries to discover apicoplast-specific compounds. Overall, we believe that the PfMev represents a user-friendly platform to investigate essential processes of the apicoplast organelle.

## Methods

### *P. falciparum* culture and maintenance

Unless otherwise noted, blood-stage *P. falciparum* parasites were cultured in human erythrocytes at 1% hematocrit in a 10mL total volume of CMA (Complete Medium with Albumax) containing RPMI 1640 medium with L-glutamine (USBiological Life Sciences), supplemented with 25mM HEPES, 0.2% sodium bicarbonate, 12.5μg/ml hypoxanthine, 5g/L Albumax II (Life Technologies) and 25μg/mL gentamicin. Cultures were maintained in $25cm^2$ gassed flasks (94% $N_2$, 3% $O_2$, 3% $CO_2$) and incubated at 37˚C.

### Construction of the p15-Mev-aSFG mevalonate bypass plasmid

The mevalonate bypass plasmid (**S1B Fig**) used a modified pACYC replicon, with a high copy number p15a origin (p15HC) and chloramphenicol resistance cassette. The sequence encoding the MVA pathway enzymes itself was developed through the generation of synthetic genes encoding *Streptococcus pneumoniae* mevalonate kinase (MevK), *Homo sapiens* phosphomevalonate kinase (PMevK), *S. pneumoniae* mevalonate 5-diphosphate decarboxylase (MVD), as well as the *E. coli* isopentenyl diphosphate isomerase (Idi). All genes were codon optimized for expression in *E. coli* and manually modified to avoid rare codons for expression in *P. falciparum*. The four genes are connected by nucleotides encoding seven amino acid flexible linkers (**S1C Fig**).

A key consideration in choosing the source of the enzymes to include in the mevalonate pathway was the availability of biochemical and structural information. The design of the four-enzyme system involved separating each protein with either a viral 2A skip peptide (to generate individual proteins) or flexible linkers (to generate a multi-domain fusion protein). In either case, it was critical to choose enzymes that could accommodate a carboxy-terminal extension. Another consideration was the size of the proteins. The yeast mevalonate enzymes

have been used extensively in synthetic biology, but are relatively large and exhibit negative feedback inhibition. Size was a driving consideration in choosing human PMevK since it is less than half the size of the yeast congener. As shown in **Fig 1**, we were able to create a chimeric four-enzyme lower mevalonate pathway fusion protein. To our knowledge, mutli-domain mevalonate pathway enzymes do not exist in nature, but may benefit from enhanced local concentrations of substrates.

To visualize apicoplast morphology another cassette was added to the plasmid expressing the first 55 amino acids of the *P. falciparum* Acyl Carrier Protein (ACP) fused to the N-terminus of a codon-optimized variant of GFP called Super Folder Green (SFG). To drive expression of the apicoplast trafficked SFG (api-SFG) and MVA bypass pathway cassettes a strong bidirectional promoter (Pcam/Psti) was used [61,62]. This plasmid, p15-Mev-aSFG, was deposited in GenBank with accession number MN822298.

### Generation of the PfMev bypass line

For generation of the PfMev line, *P. falciparum* NF54[attB] parasites [33] were transfected using methods previously described [32,63]. Briefly, 400μL of red blood cells were electroporated with 75μg each of the p15-Mev-aSFG and pINT (encoding the mycobacteriophage Bxb1 integrase) plasmids [32], after which ~2.5mL of a ~10% parasitemia culture, synchronized as schizonts, was added. After 48 hours, integration was selected by adding 25μM fosmidomycin (50x $IC_{50}$) supplemented and 50μM mevalonate. Infected red blood cells (iRBC) were first observed approximately 15–20 days after beginning drug selection. Once parasites were observed the culture was maintained on CMA medium. It should be noted that this and all subsequent experiments using mevalonate supplementation used mevalonolactone (Sigma-Aldrich, cat. #M4667), which is presumably hydrolyzed in solution to form mevalonate.

### Parasite growth curve determination

To test for the ability of PfMev parasites to use the mevalonate pathway, PfMev parasites were seeded in a 96-well plate (Corning) at a 0.5% starting parasitemia at 2% hematocrit, at a total volume of 250μL, in quadruplicate for each condition tested. Parasites were grown in the presence of 25μM fosmidomycin, and supplemented with mevalonate at concentrations of either 0μM, 2.5μM, 10μM, 50μM, or 100μM, with parasites grown in CMA medium as a control. Plates were incubated in chambers gassed with 94% $N_2$, 3% $O_2$, 3% $CO_2$ at 37˚C. The media was changed daily, and parasite samples were collected for SYBR Green staining and analysis via flow cytometry every 24 hours for four days.

To determine the amount of mevalonate required for the engineered bypass pathway, apicoplast negative parasites were generated by treating PfMev parasites with 100nM azithromycin in the presence of 50μM mevalonate for one week. Loss of the organelle was confirmed via PCR as outlined below, and additionally verified via live epifluorescence microscopy using the methods also described below. Parasite growth was determined as outlined above. Apicoplast-negative PfMev parasites were washed with 10mL CMA three times to remove mevalonate, and then grown with mevalonate at concentrations of either 0μM, 2.5μM, 10μM, 50μM, or 100μM. In parallel, PfMev parasites possessing an intact apicoplast were cultured in either 0μM or 50μM mevalonate as a control. To test if DXPR is an essential gene, PfMev Δ*dxpr* parasites were washed with CMA, and then cultured for four days in the presence (50μM) or absence of mevalonate.

For growth curve determination, parasites were stained with SYBR Green, and analyzed via flow cytometry. Parasites were collected every 24 hours for determination of parasitemia. Samples collected on days 1–3 were diluted 1:10 in PBS and stored in a 96-well plate at 4˚C. On

day 4, parasites were stained with SYBR Green by transferring 1μL of parasite culture, or 10μL of the 1:10 dilutions, to a 96-well plate containing 100μL of 1x SYBR Green in PBS, and incubated for 30 minutes while shaking. Post-incubation, 150μL of PBS was added to each well to dilute unbound SYBR Green dye. Samples were analyzed with an Attune Nxt Flow Cytometer (Thermo Fisher Scientific), with a 50μL acquisition volume, and a running speed of 25μL/minute with 10,000 total events collected.

## Confirmation of apicoplast loss

The apicoplast organellar genome was detected using PCR using primers specific for *sufB* (PF3D7_API04700). To serve as a control, primers were also used to amplify genes from the nuclear (*ldh*; PF3D7_1324900) and mitochondrial (*cox1;* mal_mito_2) genomes (**S1 Table**). 1μL of the parasite sample was added to the 50μL total volume of the PCR reaction. The parental PfMev line was used as a positive control for apicoplast detection.

## Live cell epifluorescence microscopy

For sample preparation, 100μL of parasite culture was obtained. Parasites were incubated with 30nM MitoTracker CMX-Ros (Invitrogen) and 1μg/mL 4′, 6-diamidino-2-phenylindole (DAPI) for 30 minutes at 37˚C. Cells were then washed three times with 100μL of CMA media and incubated for 5 minutes at 37˚C after each wash. Cells were resuspended in 20μL of CMA and then pipetted onto slides and sealed with wax for observation on a Zeiss AxioImager M2 microscope. A series of images spanning 5μm were acquired with 0.2μm spacing and images were deconvolved using VOLOCITY software (PerkinElmer) to report a single image in the z-plane.

## PfMev parasite immunofluorescence (IFA) microscopy

PfMev parasites were prepared for IFA as previously described [64]. Briefly, parasites were resuspended in a 1:1 solution of 4% paraformaldehyde and 0.0075% glutaraldehyde in PBS for 30 minutes to fix the cells. Cells were permeabilized for 10 minutes with 1% Triton X-100 and then reduced for 10 minutes with 0.1g/L NaBH$_4$ in PBS. Cells were blocked for 2 hours in a solution of 3% (w/v) BSA in PBS, washed and then incubated at 4˚C overnight with anti-ACP rat antibodies (1:500) [65] and rabbit anti-GFP antibodies (1:500) [66]. Cells were washed three times with PBS, and then incubated for 2 hours in a PBS solution containing 3% BSA and 1:1000 anti-rabbit Alexa Fluor 594 (Life Technologies) and 1:300 anti-rat Alexa Fluor 488 (Life Technologies). Cells were washed with PBS three times and sealed with Gold DAPI anti-fade (Life Technologies) under a coverslip sealed with nail polish. Slides were then viewed using a Zeiss AxioImager M2 microscope. A series of images spanning 5μm were acquired with 0.2μm spacing and images were deconvolved with VOLOCITY software (PerkinElmer) to report a single image in the z-plane.

## [2-$^{13}$C]-mevalonate parasite labeling and detection via LC-MS

A PfMev culture was synchronized through magnetic purification [67] and used to seed two identical 20mL cultures at 2% hematocrit and ~2% parasitemia. Parasites were cultured in media containing 25μM fosmidomycin (50x IC$_{50}$) and supplemented with either 50μM unlabeled mevalonolactone (Sigma-Aldrich, cat. #M4667) or 50μM [2-$^{13}$C]-mevalonate (Sigma-Aldrich, cat. #486604), and incubated for 48 hours, equivalent to one parasite lifecycle, to allow for incorporation of the label. After 48 hours, the parasitemia had reached ~9% and the

parasites were isolated via magnetic purification, resuspended in 12mL CMA, and quantified using a hemocytometer, with ~8x10$^7$ parasites purified from each culture.

Metabolite extraction was conducted following a previously established protocol [68]. Briefly, the parasites were pelleted, snap frozen on dry ice, and stored at -80˚C. For metabolite extraction, 100% methanol cooled on dry ice was added, and incubated on dry ice for 15 minutes with intermittent vortexing. The tube was then spun down at 4˚C and the supernatant was transferred to a separate tube and stored on dry ice. To the original tube containing the pellet, 600μL of ice cold 80% methanol/20% $H_2O$ was added, and the pellet was broken up by pipetting up and down with intermittent vortexing. The tube was then tip sonicated (Vibra-Cell, Sonics & Materials Inc.) on ice for 30 seconds with 0.5 second on/off cycles at 20% amplitude. This tube was then spun down at 4˚C and the supernatant was then pooled with 100% methanol extraction, mixed, and dried down under nitrogen gas until fully evaporated and stored at -80˚C. On the day of analysis, dried samples were resuspended in 100μL of ice cold solvent A (2mM ammonium phosphate and 2mM hexylamine in $H_2O$) and then incubated on ice for ~1.5 hours with intermittent vortexing. The sample was then spun down at 4˚C, and the supernatant was transferred to a fresh tube for metabolite analysis.

Metabolites were detected using a Dionex Ultimate 3000 uHPLC system coupled to a TSQ Vantage Triple Stage Quadrupole (QQQ) mass spectrometer (Thermo Fisher Scientific) using a Hypersil-GOLD C18 column (3μm, 100mm x 1mm, Thermo Fischer Scientific) at a flow rate of 0.4mL/min. Solvent A is listed above and Solvent B was 100% acetonitrile. The following gradient was used: 0–2 minutes: at 9% B, 2–15.5 minutes linearly increasing from 9–100% B, 15.6–17.6 minutes at 100% B, and 17.7–31.7 minutes at 9% B.

To optimize detection of IPP, a chemical standard was analyzed by injecting 5μL of 5μM IPP (Sigma) in 90% mobile phase A, and 10% mobile phase B (100% acetonitrile) (**S4 Fig**). The detection method relied on selected reaction monitoring (SRM) in negative ion mode to monitor the transition (m/z 244.9 → 79.0), in which the parent ion (244.9m/z) was selected for and fragmented, and the daughter ion fragment was detected (79.0m/z), indicative of the loss and detection of phosphate from IPP [69]. The retention time of the IPP standard was found to be ~11.9 seconds. For sample analysis, 40μL of sample was injected. In order to detect IPP we monitored the above-mentioned transition (m/z 244.9 → 79.0). However, parasites supplemented with [$^{13}$C]-mevalonate should produce [$^{13}$C]-labeled IPP, shifting the molecular mass by one Dalton to (m/z) 245.9. Thus, for samples extracted from parasite cultures supplemented with unlabeled mevalonate or [$^{13}$C]-mevalonate both transitions were monitored (m/z 244.9 → 79.0) and (m/z 245.9 → 79.0). The same SRM technique was used to look for downstream isoprenoid products, specifically farnesyl pyrophosphate (FPP). Using previously published methods, we monitored the transition (m/z 381.0 → 79) for unlabeled FPP [69]. Since FPP is formed from three isoprenoid precursors, the mass of labeled FPP should be shifted by three Daltons to (m/z) 384.0. Thus, we also monitored both transitions (m/z 381.0 → 79) and (m/z 384.0 → 79) in the labeled and unlabeled samples.

## Generation of the DXPR gene deletion constructs

DXPR was deleted through Cas9 mediated gene editing, using versions of the pL6-eGFP and the pUF1-Cas9 plasmids, which were generously provided by Dr. Jose-Juan Lopez-Rubio [38]. The pL6-eGFP plasmid was modified to facilitate plasmid construction with primers listed in **S1 Table**. The plasmid was first digested with EcoRI and NcoI, and an adaptamer (pL.3LIC. NgoMIV.F and pL.3LIC.NgoMIV.R) was inserted to create ligation independent cloning (LIC) site containing a NgoMIV cut site at the 3' end of the hDHFR drug resistance cassette. This plasmid was then cut with NotI and AflII and an adaptamer (pL.5LIC. NotI.F and

pL.5LIC. NotI.R) was inserted to create a LIC site containing a NotI cut site at the 5' end of the hDHFR drug resistance cassette. The LacO/Z fragment present in the backbone of the plasmid was removed with SapI and AflII followed by the insertion of an adaptamer (NoPro.Adapt.F and NoPro.Adapt.R). This plasmid was then cut with BglI and BsaI and an adaptamer (pL8. xBsaI.F and pL8.xBsaI.R) was inserted to change the BsaI site in the AmpR *E. coli* drug resistance cassette. The resulting plasmid was then cut with BtgZI, and a synthetic LacZ expression cassette (**S1 Table** and **S5 Fig**) flanked by two BsaI sites was inserted into the plasmid to allow for blue/white colony screening. We are calling the resulting plasmid pRS-LacZ (GenBank MN822297).

For the generation of the DXPR pRS deletion plasmid, homology arms of ~300-600bp for DXPR were amplified using the homology arm (HA) 1 and 2 forward and reverse primers (**S1 Table**) from blood stage *P. falciparum* NF54^attB genomic DNA. HA1 and HA2 primers were designed to contain ~15bp overhangs for insertion into the cut pRS plasmid using LIC methods (In-Fusion, Clontech). The pRS plasmid was digested with NotI for insertion of HA1, and with NgoMIV for HA2. The pRS plasmid also contains a guide RNA (gRNA) expression cassette. For insertion of the gRNA sequence, the plasmid was digested with BsaI (**S5 Fig**). The guide RNA was inserted as an adaptamer (**S1 Table**) into pRS using In-Fusion.

### *P. falciparum* transfections for gene deletion

For deletion of DXPR, 400μL of red blood cells were electroporated with 75μg each of pRS-DXPR and the Cas9 expression plasmid. The electroporated RBCs were mixed with ~2.5mL PfMev parasites synchronized as schizonts and cultured in CMA with 50μM mevalonate. After 48 hours, transfectants were selected with 1.5μM DSM1, 2.5nM WR99210, and 50μM mevalonate for seven days, after which they were cultured in CMA containing 50μM mevalonate. Infected red blood cells were first observed between 17 and 30 days after beginning drug selection. Once parasites were observed, the medium was switched to, and maintained on medium containing 2.5nM WR99210 and 50μM mevalonate.

### Confirmation of knockout genotype

In order to detect recombinant parasites, primers were designed to screen for integration and gene disruption at the Δ5' (5'F/ pL8 HA1 R) and Δ3' (pL8 HA2F/3'R) ends in addition to 5' (5'F/5' WT R) and 3' (3' WT F/3'R) regions of the WT gene (**S1 Table**). The same reactions were also performed on the parental PfMev line concurrently as a control. For the PCR reaction, 1μL of the parasite sample was added to each of the reactions with a 50μL total volume.

### Parasite culture, RBC purification and sample collection for transcriptomic and metabolomic analysis

Apicoplast negative parasites were generated from the PfMev parasite line, with loss of the apicoplast induced and confirmed as described above. The apicoplast negative PfMev parasites were cultured as described above, but at 2% hematocrit and supplemented with 50μM mevalonate. To remove contaminating white blood cells (WBCs), whole blood was centrifuged to remove the buffy coat and resuspended in RPMI-1640 medium twice. The hematocrit was then adjusted to 10% and the cells were passed through a neonatal high efficiency leukocyte reduction filter (Haemonetics). After removal of WBCs, the RBCs were resuspended in RPMI-1640 and adjusted to a hematocrit of 50%.

Parasites were collected for transcriptomic and metaoblomic analysis using methods similar to those previously published [70]. To generate synchronized parasites, cultures were magnetically purified through a XS MACS column (Miltenyi Biotec, Auburn CA) with a homemade

magnet (field strength of 0.93T), every 38–48 hours for four days prior to the experiment. Parasitemia and synchronicity were monitored via Giemsa stain. Parasites were also screened for mycoplasma and confirmed to be apicoplast-disrupted immediately prior to the experiment (S1 Table, S6 Fig). To initiate the experiment, a 300mL synchronized parasite culture was passed through a XS column, and used to seed 75mL flasks in quadruplicate at 2% hematocrit and 7% parasitemia. Similar uninfected cultures were set up from the same RBCs and growth medium. Parasite cultures were observed via Giemsa stain until 0–2 hours post merozoite invasion of RBCs, at which point the culture media was replaced with fresh media, corresponding to time 0 for this experiment. Samples were collected at times 0, 8, 16, 24, 36, 40 and 48 hours. For each time point 7.2mL of the parasite culture was collected, spun down, and aspirated, with 50μL of the pellet transferred to a fresh tube and immediately flash-frozen in an ethanol/dry-ice bath followed by storage at -80˚C. Samples were analyzed at the Johns Hopkins Genomic Analysis and Sequencing Core Facility using Agilent microarray chip AMADID 037237 [71]. Data can be accessed at the NCBI Gene Expression Omnibus (GSE136169). For metabolite analysis, 15mL of the parasite cultures and the uninfected RBC cultures were collected, spun down, and aspirated, with 100μL of the pellets transferred to fresh tubes and immediately flash-frozen in an ethanol/dry-ice bath followed by storage at -80˚C. Quadruplicate samples were sent to Metabolon (Morrisville, NC) for metabolite analysis.

## Supporting information

**S1 Table. Primer sequences used in this study.**
(DOCX)

**S1 Fig. Map of the transfection plasmid used to generate PfMev parasites. A)** Features of the p15a high copy (p15HC) origin of replication are shown. Sigma factor binding sites (-10 box, green, and -35 box, orange) are shown for bacterial promoters driving production of the replication primer RNA (blue) and RNA I (red), a structural RNA responsible for plasmid copy number and compatibility group. The mutation (T→G) at position -478 (relative to the origin) that increases plasmid copy number is indicated with a yellow star. **B)** The pACYC replicon required for plasmid maintenance in *E. coli* is shown, including the p15HC origin of replication and chloramphenicol acyltransferase gene (ChlorR). Transcriptional terminators are colored in red and promoters are colored in green. Parasite promoters Pcam and Psti (green arrows) drive the production of apicoplast-targeted SFG (aSFG) and the MVA pathway enzymes (MevK, PMevK, Idi and MVD). A recombination site for mycobacteriophage Bxb1 integrase is shown in purple (attP). **C)** Features of the mevalonate expression cassette are shown including the four enzymatic domains and the linker regions between them. Endonuclease sites are highlighted in orange text and the stop codon is colored red.
(TIF)

**S2 Fig. Diagram showing the PCR reactions used to confirm gene knockout parasites.** Diagram showing how the diagnostic primers listed in S1 Table were used to demonstrate integration of the hDHFR drug resistance cassette.
(TIF)

**S3 Fig. Transcriptional profiles of nuclear-encoded apicoplast genes.** The 96 genes shown in Fig 7C can be divided into four major groups based on the hierarchical clustering of their transcriptional profiles over the 48-hour life cycle. These clusters highlight the similarities in gene expression between the control data and the data collected from apicoplast-disrupted parasites (Apicoplast-).
(TIF)

**S4 Fig. LC-MS/MS detection of IPP for an analytical standard.** LC-MS/MS analysis of the pure compound IPP for use as an analytical standard was obtained through the injection of 5μL of 5μM IPP (Sigma). SRM in negative ion mode monitored the transitions m/z 244.9 → 79.0 and m/z 245.9 → 79.0 to select for the parent ion of unlabeled IPP or IPP containing a [$^{13}$C] isotope, respectively, with detection of the daughter ion. As expected, we detected both transitions due to the ~1.1% natural abundance of [$^{13}$C]. Integrating the curves, we found that the area corresponding to the m/z 245.9 → 79.0 transition was ~6.3% of the m/z 244.9 → 79.0 transition. Importantly, the retention time of the analytical standard (~11.9 seconds) corresponds to the signal detected when analyzing parasite samples.
(TIF)

**S5 Fig. Plasmid features of pRS-LacZ.** The plasmid map is shown for pRS-LacZ indicating the locations of the homology arm insertion sites (orange) and the insertion site for gRNA adaptamers (yellow). The features and sequence of the chimeric LacZα expression cassette used for Blue/White screening during insertion of the gRNA sequence are also shown.
(TIF)

**S6 Fig. Transcriptomic analysis of apicoplast-disrupted parasites. A)** Lane 1 shows a PCR product generated using primers Myco16S.F and Myco16S.R with a positive control sample of *Mycoplasma arginini*. Lanes 2 and 3 show that the uninfected RBCs (uRBCs) and apicoplast-disrupted (Api-) parasite cultures used for transcriptomic and metabolomic experiments are not infected with mycoplasma. **B)** Confirmatory PCR products showing that control parasite cultures contain genes from the nuclear (N), apicoplast (A) and mitochondrial (M) genomes while the apicoplast-disrupted (Api-) parasites cultures used for transcriptomic and metabolomic experiments no longer produce a product for the apicoplast genome. Uninfected RBCs (uRBC) were used to show that PCR products are not formed in the absence of parasites.
(TIF)

## Acknowledgments

We thank David Fidock for providing the NF54$^{attB}$ strain of *P. falciparum* parasites.

## Author Contributions

**Conceptualization:** Russell P. Swift, Sean T. Prigge.

**Data curation:** Amanda Dziedzic, Anne E. Jedlicka, Shivendra G. Tewari, Anders Wallqvist.

**Formal analysis:** Russell P. Swift, Krithika Rajaram, Amanda Dziedzic, Anders Wallqvist, Sean T. Prigge.

**Funding acquisition:** Sean T. Prigge.

**Investigation:** Russell P. Swift, Krithika Rajaram, Hans B. Liu, Amanda Dziedzic, Anne E. Jedlicka, Aleah D. Roberts, Krista A. Matthews, Hugo Jhun, Shivendra G. Tewari, Sean T. Prigge.

**Methodology:** Hans B. Liu, Namandje N. Bumpus, Anders Wallqvist.

**Project administration:** Sean T. Prigge.

**Resources:** Namandje N. Bumpus.

**Software:** Shivendra G. Tewari.

**Supervision:** Anne E. Jedlicka, Anders Wallqvist, Sean T. Prigge.

**Visualization:** Shivendra G. Tewari.

**Writing – original draft:** Russell P. Swift, Sean T. Prigge.

**Writing – review & editing:** Russell P. Swift, Krithika Rajaram, Hans B. Liu, Shivendra G. Tewari, Sean T. Prigge.

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
