## [Decision Letter · Decision Letter 0]

16 Sep 2019

Dear Dr. Prigge:

Thank you very much for submitting your manuscript "A mevalonate bypass system facilitates elucidation of plastid biology in malaria parasites" (PPATHOGENS-D-19-01463) for review by PLOS Pathogens. Your manuscript was fully evaluated at the editorial level and by independent peer reviewers. All three reviewers found your topic compelling and approach innovative, but identified some aspects of the manuscript that should be improved.

We therefore ask you to modify the manuscript according to the review recommendations before we can consider your manuscript for acceptance. Your revisions should address the specific points made by each reviewer. In particular, please note the need to include three or more replicates for all experimental findings.

(1) A letter containing a detailed list of your responses to the review comments and a description of the changes you have made in the manuscript. Please note while forming your response, if your article is accepted, you may have the opportunity to make the peer review history publicly available. The record will include editor decision letters (with reviews) and your responses to reviewer comments. If eligible, we will contact you to opt in or out.

(2) Two versions of the manuscript: one with either highlights or tracked changes denoting where the text has been changed; the other a clean version (uploaded as the manuscript file).

We hope to receive your revised manuscript within 60 days or less. If you anticipate any delay in its return, we ask that you let us know the expected resubmission date by replying to this email.

[LINK]

Sincerely,

Audrey Ragan Odom John, MD, PhD

Pearls Editor and guest Associate Editor

PLOS Pathogens

Kami Kim

Section Editor

PLOS Pathogens

Kasturi Haldar

Editor-in-Chief

PLOS Pathogens

orcid.org/0000-0001-5065-158X

Grant McFadden

Editor-in-Chief

PLOS Pathogens

orcid.org/0000-0002-2556-3526

Reviewer's Responses to Questions

**Part I - Summary**

Reviewer #1: Swift et al have generated an alternative tool for IPP rescue by genetic engineering parasites to utilize the lower mevalonate pathway. The major strengths of this study are the successful metabolic engineering of Plasmodium parasites and thorough characterization of the engineered strain. This is a tricky bit of genetic engineering in a difficult organism, including optimization and fusing of pathway genes. An easily-addressable weakness is that the authors do not clearly state the advantages and disadvantages of this alternative compared to the current tool, IPP rescue, making it difficult to assess significance. The main advantage of PfMev the authors discuss is the lower cost of the rescue compound, but this comes at the disadvantage of additional genetic manipulation of parasites which itself is costly in time and resources. Several of the applications (i.e. genetic transfection, compound screening, and forward genetic screens) mentioned by the authors have been performed using IPP rescue, suggesting that IPP rescue is sufficient to “elucidate plastid biology” for these types of experiments and not “prohibitive.” Unfortunately, the previous literature is not sufficiently discussed or cited in the manuscript and therefore do not allow for a fair evaluation of how PfMev improves upon the existing tool. I agree with authors that the application where IPP rescue is indeed prohibitive is metabolic labeling because of lack of availability of isotope-labeled reagents. Although not necessary to validate this tool, it would increase the significance of PfMev if more proof-of-concept for identification of isoprenoid products could be included. In addition, this engineering is halfway to a bypass in P. berghei for which chemical supplementation cannot easily be performed in mice; this would indeed open up many studies of plastid biology in this in vivo model. In terms of significance, metabolic labeling of isoprenoid products and facilitating study of apicoplast biology in in vivo malaria models are what I find most exciting for this new tool. This work should certainly be shared with the community, but I would recommend a more balanced presentation about its utility compared to the existing tool of IPP rescue.

Reviewer #2: The manuscript by Swift et al. describes an elegantly conceptualized and executed body of work that is likely to provide an excellent tool to understand plastid biology in Plasmodium falciparum. They have generated transgenic parasites, called PfMev line, expressing four enzymes able to utilize mavelonate to synthesize isopentenyl pyrophosphate (IPP). These enzymes were judiciously chosen from three different organisms and were physically fused to generate an artificial multifunctional enzyme complex, which then was expressed through a p15a origin of replication containing vector integrated at a nonessential chromosomal locus. These transgenic parasites were shown to become independent of apicoplast localized IPP synthesis as long as mavelonate was provided in the medium. These parasites could be made to lose their apicoplast when treated with azithromycin. The transcriptomic and metabolomic analysis of these apicoplast-minus parasites revealed little alterations: transcription of genes encoding apicoplast-targeted proteins was unaltered as were most of the metabolites. They were also able to ablate an essential gene involved in apicoplast-localized IPP synthesis in PfMev parasites. The manuscript is well written, experiments provide clear results, and the findings have far-reaching implications. There are a few minor points that I would suggest to improve the manuscript:

1. Authors do not provide rationale as to of how they came up to choose the four enzymes from 3 species to generate their bypass system. It would be important to describe this in the Results section. One has to go to the Methods section to even find out that the genes were different species. The rationale for their choices would be instructive to readers even outside the parasitology field.

2. They should provide a fully annotated DNA sequence of the plasmid used to generate PfMev parasites. There are many interesting and important aspects of this novel vector that are not apparent from the Supplementary Fig. 1, and a deposit in GenBank is not the same as a color-coded DNA sequence indicating various features of the plasmid included in the Supplementary information.

3. They do see some of the metabolites that seem to be altered in PfMev line . It would help to provide some information about them in the Results text in addition to being listed in Fig. 8.

4. It would help to provide a structure of mavelonolectone in Fig. 2 and state that this is what was used to supplement the culture medium in the Results section.

5. It would be worthwhile to point out that one possible reason for the lack of apicoplast biosynthetic activity in apicoplast-minus parasites, in spite of having most of the apicoplast-directed proteins in fragmented vesicles, could be the absence of a chaperone encoded by the apicoplast genome. This could render the targeted proteins unable to form functional enzymes.

Reviewer #3: This is a very well-written manuscript describing the generation, and characterisation, of a P. falciparum parasite line that can replicate normally after the apicoplast has been disrupted provided that mevalonate is present in the culture medium. This parasite line can synthesise isoprenoid precursors via an exogenous, mevalonate-dependent pathway, bypassing the parasite’s endogenous methylerythritol phosphate pathway. In characterising the parasite line, the authors were able to delete a normally-essential enzyme (the target of fosmidomycin). The authors also carried out metabolomic and transcriptomic studies to investigate the consequences for the parasite of apicoplast disruption, generating valuable new information. I believe the manuscript will be of high interest to the readership of PLOS Pathogens and the parasite line the authors have generated will be in demand by a number of malaria research labs around the world. I do not have any major criticism of the work but have identified a number of minor issues which I hope the authors will address. These are listed in order of appearance below.

**Part II – Major Issues: Key Experiments Required for Acceptance**

Reviewer #1: No additional experiments are required to validate the study conclusions that the lower MVA pathway has been engineered into a P. falciparum strain generating PfMev.

Reviewer #2: (No Response)

Reviewer #3: No new experiments are required, but I recommend that some of the experiments are repeated once more to generate averages from three independent experiments (see details below).

**Part III – Minor Issues: Editorial and Data Presentation Modifications**

Reviewer #1: In Introduction, establishing what is possible with IPP rescue prior to this manuscript, key citations are missing:

Use of IPP for drug screens: numerous, including Bowman et al. Anti-apicoplast and gametocytocidal screening to identify the mechanisms of action of compounds within the Malaria Box, Uddin et al. Validation of Putative Apicoplast-Targeting Drugs Using a Chemical Supplementation Assay in Cultured Human Malaria Parasites, Gisselberg et al. Specific Inhibition of the Bifunctional Farnesyl/Geranylgeranyl Diphosphate Synthase in Malaria Parasites via a New Small-Molecule Binding Site

Use of IPP for making gene knockouts and grow otherwise non-viable transfectants: Florentin et al. The Clp proteolytic system acts as an essential nexus for Plasmodium plastid biogenesis (https://www.biorxiv.org/content/10.1101/718452v1.full), Gisselberg et al. The suf iron-sulfur cluster synthesis pathway is required for apicoplast maintenance in malaria parasites

Use of IPP for forward genetic screens: Tang et al. A mutagenesis screen for essential plastid biogenesis genes in human malaria parasites (https://doi.org/10.1371/journal.pbio.3000136)

In Results, Design and validation of a mevalonate dependent isoprenoid synthesis pathway, the authors mention the need for IPP isomerase (idi) for proper balancing of IPP/DMAPP levels in E. coli. However, there is a parasite-encoded IPP isomerase annotated and no additional isomerase is required for IPP rescue. It seems the idi may be superfluous in the parasite. Was this tested?

In Results, Transcriptomic analysis of apicoplast disrupted parasites, the continued expression and detection of nuclear-encoded apicoplast proteins upon loss of the apicoplast under IPP rescue has already been demonstrated in previous papers. These should be cited, since the literature already indicated that expression of nuclear-encoded apicoplast proteins is not affected by apicoplast loss. The advance of this study is to show evidence for expression of many nuclear-encoded apicoplast genes using transcriptomic analysis.

Was there any attempt to reconstitute the upper MVA pathway (so no chemical supplementation is required)? Please discuss in context of potential for use in P berghei where chemical supplementation is difficult or not possible. This would increase the significance of this work.

To show utility of strain for identification of isoprenoid products, show evidence of labeled final products, for example the volatile terpenoid compounds described in Kelly et al. This would be very exciting and increase the significance of this work.

Fig 5. Please show the mass spectrum of the products compared to chemical standards to validate the peaks are correctly identified.

Reviewer #2: (No Response)

Reviewer #3: Minor issues.

1) In the “author summary”, IPP is referred to as “relatively unstable”. This term is too vague. Relative to what?

2) In several places the authors refer to “as little as 10 µM”. I would suggest that the authors refrain from using terms such as this, or explain why they consider 10 µM to be “little”. I assume they are comparing it to how much IPP has been used in the past to allow disruption of the apicoplast, but this is not clear (and the comparison unnecessary).

3) The last paragraph of the Discussion is a bit repetitive of other parts of the manuscript and can be shortened.

4) In the legend of Figure 2, the authors should clarify that the following sentence refers to parasite treated with fosmidomycin: “PfMev parasites not supplemented with mevalonate failed to grow”.

5) The data shown in Figures 2 and 4 are from two independent experiments and the error bars shown are SEM. SEM should not be used if the experiment has only been carried out twice. I would encourage the authors to repeat these experiments a third time.

6) Legends of Figures 3, 4 and 6. The following sentence should be modified: “Images are 10 microns long by 10 microns wide”. The actual images are much bigger than that (what the authors mean is that the size of the image can be used to represent 10 microns).

7) Legend of Figure 5. To be consistent, the authors should avoid using [12C]-mevalonate and refer to it as unlabeled mevalonate.

8) Figure 6. There is no indication of the number of independent experiments that were carried out to generate the data shown in part D. The legend also does not indicate what the error bars represent.

9) Figure 8 legend. The word “the” needs to be inserted in the following sentence: “The hierarchal clustering method is THE same as described…”

10) The references needs to be carefully checked to italicise species names. Reference 32 has errors within the title. Reference 47 has a minor error next to the citation details.

11) Figures 4, 6 and S4. Include units for 1000 and 500.

PLOS authors have the option to publish the peer review history of their article (what does this mean?). If published, this will include your full peer review and any attached files.

Reviewer #1: No

Reviewer #2: No

Reviewer #3: Yes: Prof Kevin J Saliba

---

## [Editor Report · Decision Letter 1]

10 Jan 2020

Dear Dr. Prigge,

We are pleased to inform that your manuscript, "A mevalonate bypass system facilitates elucidation of plastid biology in malaria parasites", has been editorially accepted for publication at PLOS Pathogens. 

Before your manuscript can be formally accepted and sent to production, you will need to complete our formatting changes, which you will receive by email within a week. Please note that your manuscript will not be scheduled for publication until you have made the required changes.

IMPORTANT NOTES

(1) Please note, once your paper is accepted, an uncorrected proof of your manuscript will be published online ahead of the final version, unless you’ve already opted out via the online submission form. If, for any reason, you do not want an earlier version of your manuscript published online or are unsure if you have already indicated as such, please let the journal staff know immediately at plospathogens@plos.org.

(2) Copyediting and Proofreading: The corresponding author will receive a typeset proof for review, to ensure errors have not been introduced during production. Please review the PDF proof of your manuscript carefully, as this is the last chance to correct any errors. Please note that major changes, or those which affect the scientific understanding of the work, will likely cause delays to the publication date of your manuscript. 

(3) Appropriate Figure Files: Please remove all name and figure # text from your figure files. Please also take this time to check that your figures are of high resolution, which will improve the readbility of your figures and help expedite your manuscript's publication. Please note that figures must have been originally created at 300dpi or higher. Do not manually increase the resolution of your files. For instructions on how to properly obtain high quality images, please review our Figure Guidelines, with examples at: http://journals.plos.org/plospathogens/s/figures.

(4) Striking Image: Please upload a striking still image to accompany your article if one is available (you can include a new image or an existing one from within your manuscript). Should your paper be accepted, this image will be considered for our monthly issue image and may also appear on our website to feature your article. Please upload this as a separate file, selecting "striking image" as the file type upon upload. Please also include a separate "Other" file with a caption, including credits and any potential copyright information. Please do not include the caption in the main article file. If your image is from someone other than yourself, please ensure that the artist has read and agreed to the terms and conditions of the Creative Commons Attribution License at http://journals.plos.org/plospathogens/s/content-license. Please note that PLOS cannot publish copyrighted images.

(5) Press Release or Related Media: If your institution or institutions have a press office, please notify them about your upcoming paper at this point, to enable them to help maximize its impact. If they will be preparing press materials for this manuscript, please inform our press team in advance at plospathogens@plos.org as soon as possible. We ask that you contact us within one week to plan ahead of our fast Production schedule. If you need to know your paper's publication date for related media purposes, you must coordinate with our press team, and your manuscript will remain under a strict press embargo until the publication date and time. This means an early version of your manuscript will not be published ahead of your final version. 

(6)  PLOS requires an ORCID iD for all corresponding authors on papers submitted after December 6th, 2016. Please ensure that you have an ORCID iD and that it is validated in Editorial Manager.  To do this, go to ‘Update my Information’ (in the upper left-hand corner of the main menu), and click on the Fetch/Validate link next to the ORCID field.  This will take you to the ORCID site and allow you to create a new iD or authenticate a pre-existing iD in Editorial Manager

(7) Update your Profile Information: Now that your manuscript has been provisionally accepted, please log into Editorial Manager and update your profile, if needed. Go to https://www.editorialmanager.com/ppathogens, log in, and click on the "Update My Information" link at the top of the page. Please update your user information to ensure an efficient production and billing process. 

(8) LaTeX users only: Our staff will ask you to upload a TEX file in addition to the PDF before the paper can be sent to typesetting, so please carefully review our Latex Guidelines http://journals.plos.org/plospathogens/s/latex in the meantime.

(9) If you have associated protocols in protocols.io, please ensure that you make them public before publication to guarantee immediate access to the methodological details.

Best regards,

Audrey Ragan Odom John, MD, PhD

Pearls Editor

PLOS Pathogens

Kami Kim

Section Editor

PLOS Pathogens

Kasturi Haldar

Editor-in-Chief

PLOS Pathogens

orcid.org/0000-0001-5065-158X

Michael Malim

Editor-in-Chief

PLOS Pathogens

orcid.org/0000-0002-7699-2064
---

## [Editor Report · Acceptance letter]

22 Jan 2020

Dear Dr. Prigge,

We are delighted to inform you that your manuscript, "A mevalonate bypass system facilitates elucidation of plastid biology in malaria parasites," has been formally accepted for publication in PLOS Pathogens.

Best regards,

Kasturi Haldar

Editor-in-Chief

PLOS Pathogens

orcid.org/0000-0001-5065-158X

Michael Malim

Editor-in-Chief

PLOS Pathogens

orcid.org/0000-0002-7699-2064